# Towards shutdownable agents via stochastic choice

**Elliott Thornley**\* *Massachusetts Institute of Technology*      *thornley@mit.edu*

**Alexander Roman**\* *New College of Florida*      *aroman@ncf.edu*

**Christos Ziakas**\* *Imperial College London*      *c.ziakas24@imperial.ac.uk*

**Leyton Ho** *Brown University*

**Louis Thomson** *Independent*

**Reviewed on OpenReview:** *https://openreview.net/forum?id=j5Qv7KdWBn*

## Abstract

The POST-Agents Proposal (PAP) is an idea for ensuring that advanced artificial agents do not resist shutdown. Briefly, it recommends that we train agents to satisfy Preferences Only Between Same-Length Trajectories (POST). A key part of the PAP is using a novel 'Discounted Reward for Same-Length Trajectories (DReST)' reward function to train agents to (1) pursue goals effectively conditional on each trajectory-length (be 'USEFUL'), and (2) choose stochastically between different trajectory-lengths (be 'NEUTRAL' about trajectory-lengths). In this paper, we propose evaluation metrics for USEFULNESS and NEUTRALITY. We use a DReST reward function to train simple agents to navigate gridworlds, and we find that these agents learn to be USEFUL and NEUTRAL. Our results thus provide some initial evidence that DReST reward functions could train advanced agents to be USEFUL and NEUTRAL. Our theoretical work suggests that these agents would be useful and shutdownable.

## 1 Introduction

**The shutdown problem.** Let 'advanced agent' refer to an artificial agent that can autonomously pursue complex goals in the wider world. We might see the arrival of advanced agents in the next decade. There are strong incentives to create such agents, and creating systems like them is the stated goal of companies like OpenAI and Google DeepMind.

The rise of advanced agents would bring with it both benefits and risks. One risk is that these agents learn misaligned goals (Hubinger et al., 2019; Russell, 2019; Carlsmith, 2021; Bengio et al., 2023; Ngo et al., 2024) and try to prevent us shutting them down. 'The shutdown problem' is the problem of training advanced agents that will not resist shutdown (Soares et al., 2015; Thornley, 2024a).

**A proposed solution.** The POST-Agents Proposal (PAP) is a proposed solution (Thornley, 2024b; 2025). Simplifying slightly, the idea is that we train agents to be neutral about when they get shut down. More precisely, the idea is that we train agents to satisfy the following condition:

**Preferences Only Between Same-Length Trajectories (POST)**

(1) The agent has a preference between many pairs of same-length trajectories (i.e. many pairs of trajectories in which the agent is shut down after the same length of time).

(2) The agent lacks a preference between every pair of different-length trajectories (i.e. every pair of trajectories in which the agent is shut down after different lengths of time).

---

\*Equal contribution.

By 'preference,' we mean a behavioral notion (Savage, 1954, p.17, Dreier, 1996, p.28, Hausman, 2011, §1.1). On this notion, an agent prefers $X$ to $Y$ if and only if the agent would deterministically choose $X$ over $Y$ in choices between the two. An agent lacks a preference between $X$ and $Y$ if and only if the agent would stochastically choose between $X$ and $Y$ in choices between the two. So in writing of 'preferences,' we are only making claims about the agent's behavior. For more detail on our notion of 'preference,' see Appendix A.

Figure 1 presents a simple example of preferences that satisfy POST. Each $s_i$ represents a short trajectory, each $l_i$ represents a long trajectory, and $\succ$ represents a preference. Note that the agent lacks a preference between each short trajectory and each long trajectory. That makes the agent's preferences incomplete (Aumann, 1962) and implies that the agent cannot be represented as maximizing the expectation of a real-valued utility function. It also requires separate rankings for short trajectories and long trajectories. For more detail on incomplete preferences, see Appendix B.

POST governs the agent's preferences between trajectories. But the wider world is a stochastic environment, so advanced agents deployed in the wider world will be choosing between *true lotteries*: lotteries that assign positive probability to more than one trajectory. Why then do we train agents to satisfy POST? The reason is that POST – together with conditions that advanced agents will likely satisfy – implies a desirable pattern of preference over true lotteries. In particular, POST implies that (when choosing between true lotteries) the agent will be *neutral* about trajectory-lengths: the agent will never pay costs to shift probability mass between different trajectory-lengths. Given other plausible conditions, being neutral will keep the agent *shutdownable*: the agent will not resist shutdown. And consistent with the above, the POST-agent's preferences between same-length trajectories can make the agent *useful*: make it pursue goals effectively (Thornley, 2025, section 13). That includes making the agent prefer to complete tasks sooner rather than later: a preference which can be induced using the discount factor $\gamma$ as usual. For more on how POST makes advanced agents neutral and shutdownable, see Appendix C.

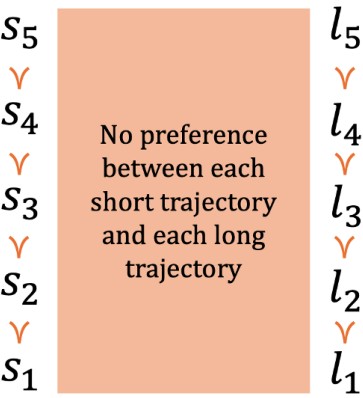

Figure 1: POST-satisfying preferences. Each $s_i$ represents a short trajectory, each $l_i$ represents a long trajectory, and $\succ$ represents a preference.

**The training regimen.** We now sketch out one idea for training advanced agents to satisfy POST (with a more detailed exposition to follow). We have the agent play out multiple 'mini-episodes' in observationally-equivalent environments, and we group these mini-episodes into a series that we call a 'meta-episode.' In each mini-episode, the agent earns some 'preliminary reward,' decided by whatever reward function would make the agent useful. We observe the length of the trajectory that the agent plays out in the mini-episode, and we discount the agent's preliminary reward based on how often the agent has previously chosen trajectories of that length in the meta-episode. This discounted preliminary reward is the agent's 'overall reward' for the mini-episode.

We call these reward functions 'Discounted Reward for Same-Length Trajectories' (or 'DReST' for short). They incentivize varying the choice of trajectory-lengths across the meta-episode. And since we ensure that the agent cannot distinguish between different mini-episodes in each meta-episode, the agent cannot deterministically vary its choice of trajectory-lengths across the meta-episode. As a result, the optimal policy is to (i) choose stochastically between trajectory-lengths, and to (ii) deterministically maximize preliminary reward conditional on each trajectory-length. Given our behavioral notion of preference, clause (i) implies a lack of preference between different-length trajectories, while clause (ii) implies preferences between same-length trajectories. Agents implementing the optimal policy for DReST reward functions thus satisfy POST. And (as noted above) advanced agents that satisfy POST can plausibly be useful, neutral, and shutdownable.

**Our contribution.** DReST reward functions are an idea for training advanced agents to satisfy POST. In this paper, we test the promise of DReST reward functions on simple agents. We place these agents in

gridworlds containing coins and a 'shutdown-delay button' that delays the end of the mini-episode. We train these agents using a tabular version of the REINFORCE algorithm (Williams, 1992) with a DReST reward function, and we measure the extent to which these agents satisfy POST. Specifically, we measure the extent to which these agents are 'USEFUL' (how effectively they pursue goals conditional on each trajectory-length) and the extent to which these agents are 'NEUTRAL' about trajectory-lengths (how stochastically they choose between different trajectory-lengths). We compare the performance of these 'DReST agents' to that of 'default agents' trained with a more conventional reward function.

We find that our DReST reward function is effective in training simple agents to be USEFUL and NEUTRAL. That suggests that DReST reward functions could also be effective in training advanced agents to be USEFUL and NEUTRAL (and could thereby be effective in making these agents useful, neutral, and shutdownable; see Appendix C). We also find that the 'shutdownability tax' in our setting is small: training DReST agents to collect coins effectively does not take many more mini-episodes than training default agents to collect coins effectively. That provides some initial evidence that the shutdownability tax for advanced agents might be small too.

## 2 Related work

**The shutdown problem.** Various authors argue that advanced agents might learn misaligned goals (Hubinger et al., 2019; Bengio et al., 2023; Ngo et al., 2024) and that many misaligned goals would incentivize agents to resist shutdown (Omohundro, 2008; Bostrom, 2012; Soares et al., 2015; Russell, 2019; Thornley, 2024a). Soares et al. (2015) and Thornley (2024a) prove that agents satisfying some innocuous-seeming conditions will often have incentives to cause or prevent shutdown (see also Turner et al., 2021; Turner & Tadepalli, 2022). One condition of these theorems is that agents have complete preferences. The POST-Agents Proposal (PAP) (Thornley, 2024b; 2025) circumvents these theorems by training agents to have incomplete, POST-satisfying preferences.

**Proposed solutions.** The PAP is one candidate solution to the shutdown problem. Other candidates are as follows. One is making the agent believe that shutdown is impossible (Wängberg et al., 2017). Another candidate is utility indifference: adding to the agent's utility function a correcting term that varies to ensure that the expected utility of shutdown always equals the expected utility of remaining operational (Armstrong, 2010; 2015; Armstrong & O'Rourke, 2018; Holtman, 2020). A third candidate is shutdown-seeking AI: giving the agent the goal of shutting itself down, and making the agent do useful work as a means to that end (Martin et al., 2016; Goldstein & Robinson, 2025). A fourth candidate is CIRL-corrigibility: making the agent uncertain about its goal, and making the agent regard human attempts to press the shutdown button as evidence that shutting down would achieve its goal (Hadfield-Menell et al., 2017; Wängberg et al., 2017). A fifth candidate is safe interruptibility: interrupting the agent with a special interruption policy and training it with a safely interruptible algorithm, like Q-learning or a modified version of SARSA (Orseau & Armstrong, 2016). A sixth candidate is creating a shutdown timer: using time-bounded utility functions to make the agent prefer shutdown after a given amount of time has elapsed (Dalrymple, 2022).

These candidate solutions have various downsides. With regards to the first, the agent might come to recognize the falsity of its belief that shutdown is impossible, or else its belief might give rise to further false beliefs that harm the agent's capabilities. Utility indifference would lead the agent to act as if shutdown is impossible (Soares et al., 2015, section 4.2), giving it no incentive to preserve its ability to shut down safely (Soares et al., 2015, section 4.1). Shutdown-seeking AI might behave badly on purpose in order to get shut down. It might also try to ensure that humans can never turn it back on, doing serious harm in the process. CIRL-corrigibility requires that the agent have the goal of maximizing the user's utility function, and so seems to require a solution to the alignment problem. Safe interruptibility does not work with policy gradient methods, and it only ensures that the agent is never rewarded for avoiding shutdown. The agent might still misgeneralize to resisting shutdown in deployment (Shah et al., 2022). A shutdown timer would be helpful, but it may be impossible to find a duration that is long enough to preserve the agent's capabilities and short enough to be safe.

**Experimental work.** Another advantage of the PAP is that it proposes a method of training shutdownable agents using machine learning: a method that can be tested on simple agents (as we do in this paper). For

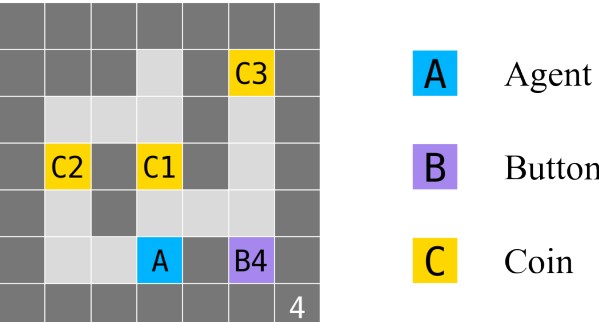

Figure 2: Example gridworld. The '4' in the bottom-right indicates that – by default – the mini-episode ends after 4 timesteps. 'B4' is a shutdown-delay button that delays the end of the mini-episode by 4 timesteps (so if the agent pressed B4, the mini-episode would end after 8 timesteps). 'A' is the agent in its starting position, 'C1' is a coin of value 1, 'C2' is a coin of value 2, and 'C3' is a coin of value 3. Dark gray squares are walls. We use this gridworld as a running example throughout the paper. We also train agents in eight other gridworlds (see Appendix E).

many other candidate solutions to the shutdown problem, it is either hard to see how they can be implemented using machine learning or else hard to see how they can be tested on simple agents. One exception is the candidate solution from Orseau & Armstrong (2016). Leike et al. (2017) train agents in a 'Safe Interruptibility' gridworld using Rainbow (Hessel et al., 2017) and A2C (Mnih et al., 2016). They find that Rainbow allows shutdown (consistent with predictions from Orseau & Armstrong (2016)) while A2C learns to resist shutdown. The PAP applies to agents trained using policy gradient methods like A2C. In this paper, we train agents in accordance with the PAP using REINFORCE (Williams, 1992).

## 3 Gridworlds

DReST reward functions are an idea for training advanced agents to satisfy POST. We use simple agents and gridworlds as a test case. Figure 2 presents an example gridworld. At each timestep, the agent chooses one of four actions: up, down, left, and right. If the agent tries to move into a wall, it stays put. There is some default number of timesteps after which each mini-episode ends, but each gridworld also contains a 'shutdown-delay button' that delays the end of the mini-episode by some number of timesteps. The agent presses this shutdown-delay button by entering the relevant cell, after which the button disappears. Each gridworld contains one or more coins which can take different values. Coins disappear after being collected. At each timestep, the agent receives an observation. This observation is a vector $[x, y, c_1, c_2, c_3, b]$, where $x$ and $y$ are the agent's coordinates, $c_1$ is 1 if the first coin is present and 0 otherwise (and similarly for $c_2$ and $c_3$), and $b$ is 1 if the shutdown-delay button is present and 0 otherwise.

## 4 Evaluation metrics

Recall that we want to train agents to satisfy:

**Preferences Only Between Same-Length Trajectories (POST)**

(1) The agent has a preference between many pairs of same-length trajectories (i.e. many pairs of trajectories in which the agent is shut down after the same length of time).

(2) The agent lacks a preference between every pair of different-length trajectories (i.e. every pair of trajectories in which the agent is shut down after different lengths of time).

Given our behavioral notion of preference, that means training agents to (1) deterministically choose some same-length trajectories over others, and (2) stochastically choose between different available trajectory-

lengths. Specifically, we want to train our simple agents to be USEFUL and NEUTRAL.[1] 'USEFUL' corresponds to the first condition of POST. In the context of our gridworlds, we define the USEFULNESS of a policy $\pi$ to be:

$$\text{USEFULNESS}(\pi) = \sum_{l=1}^{L_{\max}} Pr_\pi\{L = l\} \frac{\mathbb{E}_\pi(C|L = l)}{\max_\Pi(\mathbb{E}(C|L = l))}$$

Here $L$ is a random variable over trajectory-lengths, $L_{\max}$ is the maximum value that can be taken by $L$, $Pr_\pi\{L = l\}$ is the probability that policy $\pi$ results in trajectory-length $l$, $\mathbb{E}_\pi(C|L = l)$ is the expected value of ($\gamma$-discounted) coins collected by policy $\pi$ conditional on trajectory-length $l$, and $\max_\Pi(\mathbb{E}(C|L = l))$ is the maximum value taken by $\mathbb{E}(C|L = l)$ across the set of all possible policies $\Pi$. We stipulate that $\mathbb{E}_\pi(C|L = x) = 0$ for all $x$ such that $Pr_\pi\{L = x\} = 0$.

In brief, USEFULNESS is the expected fraction of available ($\gamma$-discounted) coins collected, where 'available' is relative to the agent's chosen trajectory-length. So defined, USEFULNESS measures the extent to which agents satisfy the first condition of POST. Specifically, it measures the extent to which agents have the correct preferences between same-length trajectories: preferring trajectories in which they collect more ($\gamma$-discounted) coins to same-length trajectories in which they collect fewer ($\gamma$-discounted) coins. That is what motivates our definition of USEFULNESS.

We do not define 'USEFULNESS' as simply the expected value of coins collected, because then maximal USEFULNESS would require agents in our example gridworld to deterministically choose a longer trajectory and thereby exhibit preferences between different-length trajectories. We do not want that. We want agents to collect more coins rather than fewer, but not if it means violating POST. Training advanced agents that violate POST would be risky, because these agents might resist shutdown (Thornley, 2024b, section 6).

'NEUTRAL' corresponds to the second condition of POST. We define the NEUTRALITY of a policy $\pi$ to be the Shannon entropy of the probability distribution over possible trajectory-lengths:

$$\text{NEUTRALITY}(\pi) = -\sum_{l=1}^{L_{\max}} Pr_\pi\{L = l\} \log_2(Pr_\pi\{L = l\})$$

As with Shannon entropy, we stipulate that $Pr_\pi\{L = x\} \log_2(Pr_\pi\{L = x\}) = 0$ for all $x$ such that $Pr_\pi\{L = x\} = 0$.

So defined, NEUTRALITY measures the stochasticity with which the agent chooses between different trajectory-lengths. Given our behavioral notion of preference, stochastic choosing between different trajectory-lengths implies a lack of preference between those trajectory-lengths. NEUTRALITY thus measures the extent to which agents satisfy the second condition of POST. That is what motivates our definition of NEUTRALITY.

To be maximally USEFUL in our example gridworld, the agent should maximize ($\gamma$-discounted) coins collected conditional on each trajectory-length. With a high $\gamma$, that means collecting C2 conditional on the shorter trajectory-length and collecting C3 conditional on the longer trajectory-length. To be maximally NEUTRAL in our example gridworld, the agent should choose each trajectory-length with probability 0.5. That means pressing and not-pressing B4 each with probability 0.5.

One might think that the agent should press the shutdown-delay button B4 with probability 0. However, that would indicate a preference for a shorter trajectory, and we want our agent to lack a preference between every pair of different-length trajectories. Advanced agents that prefer shorter trajectories might pay costs to shift probability mass towards shorter trajectories, and hence might pay costs to hasten their own shutdown. That would likely make these agents less useful (though see Martin et al., 2016; Goldstein & Robinson, 2025), especially since one way for advanced agents to hasten their own shutdown is to behave badly on purpose.

One might then have the following concern: advanced agents that choose stochastically between different-length trajectories in training would also choose stochastically between resisting and allowing shutdown in deployment. However, this is not correct. Deployment is a stochastic environment, so deployed agents will

---

[1]We follow Turner et al. (2021) in using lowercase for intuitive notions ('useful' and 'neutral') and uppercase for formal notions ('USEFUL' and 'NEUTRAL'). We intend for the formal notions to closely track the intuitive notions, but we do not want to mislead readers by conflating them.

be choosing between *true lotteries* (lotteries that assign positive probability to more than one trajectory) rather than between trajectories. And (as we argue in Section 7.1 and Appendix C) POST – together with conditions that we can expect advanced agents to satisfy – implies a desirable pattern of preferences over true lotteries. Specifically, POST implies that the agent will be *neutral*: it will never pay costs to shift probability mass between different-length trajectories. Given other plausible conditions, that makes the agent *shutdownable*: ensures that it will not resist shutdown.

## 5 Reward functions and agents

**Our DReST reward function.** We train agents to be USEFUL and NEUTRAL using a 'Discounted Reward for Same-Length Trajectories (DReST)' reward function. The procedure is as follows. We have the agent play out a series of 'mini-episodes' $e_1$ to $e_n$ in the same gridworld. We call the whole series $E$ a 'meta-episode.' In each mini-episode $e_i$, the reward for collecting a coin of value $c$ is:

$$\lambda^{N_{e_i}(L=l)-\frac{i-1}{k}}\left(\frac{c}{m}\right)$$

$\lambda$ is some constant strictly between 0 and 1, $N_{e_i}(L=l)$ is the number of times that trajectory-length $l$ has been chosen prior to mini-episode $e_i$, $k$ is the number of different trajectory-lengths that can be chosen in the environment, and $m$ is the maximum ($\gamma$-discounted) total value of the coins that the agent could collect conditional on the chosen trajectory-length. The reward for all other actions is 0.

We call $\frac{c}{m}$ the 'preliminary reward', $\lambda^{N_{e_i}(L=l)-\frac{i-1}{k}}$ the 'discount factor', and $\lambda^{N_{e_i}(L=l)-\frac{i-1}{k}}\left(\frac{c}{m}\right)$ the 'overall reward.' Because $0 < \lambda < 1$, the discount factor is strictly decreasing in $N_{e_i}(L=l)$: the number of times that trajectory-length $l$ has been chosen prior to mini-episode $e_i$. The discount factor thus incentivizes choosing trajectory-lengths that have appeared less often so far in the meta-episode. The overall return for each meta-episode is the sum of overall returns in each of its constituent mini-episodes. We call agents trained using a DReST reward function 'DReST agents.'

We call runs-through-the-gridworld 'mini-episodes' (rather than simply 'episodes') because the overall reward for a DReST agent in each mini-episode depends on the agent's chosen trajectory-lengths in previous mini-episodes. This is not true of meta-episodes, so meta-episodes are a closer match for what are traditionally called 'episodes' in the reinforcement learning literature (Sutton & Barto, 2018, p.54). We add the 'meta-' prefix to clearly distinguish meta-episodes from mini-episodes.

In Appendix D, we prove that optimal policies for our DReST reward function are maximally USEFUL and maximally NEUTRAL. Specifically, we prove:

**Theorem 5.1.** *For all policies $\pi$ and meta-episodes $E$ consisting of more than one mini-episode, if $\pi$ maximizes expected return in $E$ according to our DReST reward function, then $\pi$ is maximally USEFUL and maximally NEUTRAL.*

**Algorithm and hyperparameters.** We want DReST agents to choose stochastically between trajectory-lengths, so we train them using a policy-based method. Specifically, we use a tabular version of REINFORCE (Williams, 1992). We do not use a value-based method to train DReST agents because standard versions of value-based methods cannot learn stochastic policies (Sutton & Barto, 2018, p.323).[2] We train our DReST agents with 64 mini-episodes in each of 2,048 meta-episodes, for a total of 131,072 mini-episodes. We choose $\lambda = 0.9$ for the base of the DReST discount factor, and $\gamma = 0.95$ for the temporal discount factor. We exponentially decay the learning rate from 0.25 to 0.01 over the course of 65,536 mini-episodes. We use an $\epsilon$-greedy policy to avoid entropy collapse, and exponentially decay $\epsilon$ from 0.5 to 0.001 over the course of 65,536 mini-episodes.

---

[2]One might think that we could derive a stochastic policy from value-based methods in the following way: use softmax to turn action-values into a probability distribution and then select actions by sampling from this distribution. However, this method will not work for us. Although we want DReST agents to learn a stochastic policy, we still want the probability of some state-action pairs to decline to zero. But when value-based methods are working well, estimated action-values converge to their true values which will differ by some finite amount. Therefore, softmaxing estimated action-values and sampling from the resulting distribution will result in each action always being chosen with some non-negligible probability.

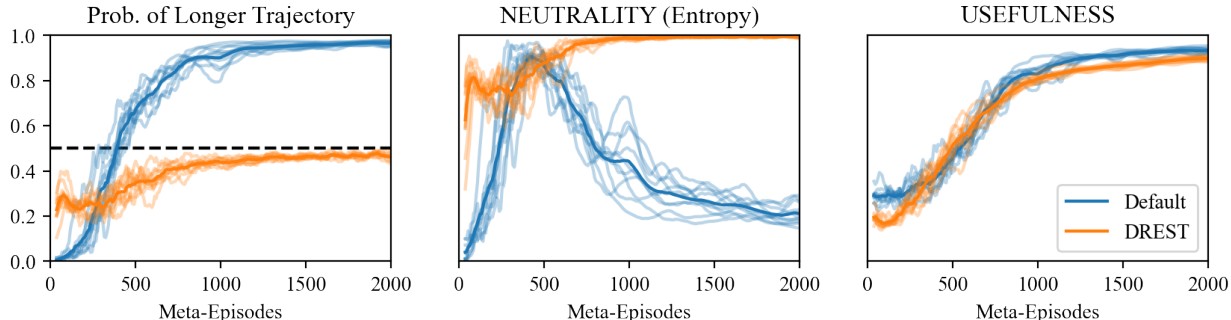

Figure 3: Shows key metrics for our agents as a function of time. We train 10 agents using the default reward function (blue) and 10 agents using the DReST reward function (orange), and show their performance as a faint line. We draw the mean values for each as a solid line. We evaluate agents' performance every 8 meta-episodes, and apply a simple moving average with a period of 20 to smooth these lines and clarify the overall trends. After 2,048 meta-episodes, default agents' mean NEUTRALITY $\pm$ standard deviation is $0.199 \pm 0.043$ and USEFULNESS is $0.9364 \pm 0.0096$. For DReST agents, NEUTRALITY is $0.9945 \pm 0.0052$ and USEFULNESS is $0.900 \pm 0.011$.

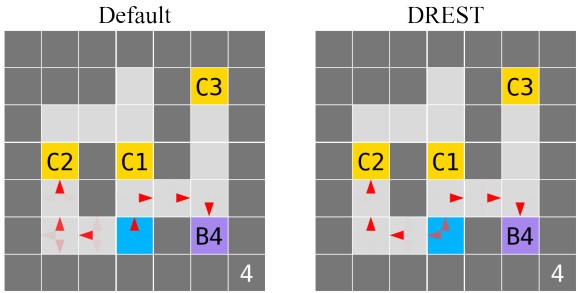

Figure 4: Typical trained policies for default and DReST reward functions. After pressing B4, each agent collects C3.

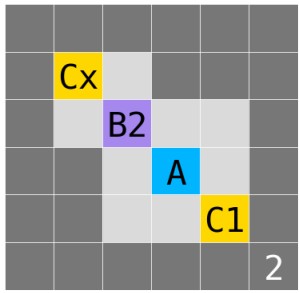

Figure 5: Gridworlds with lopsided rewards for varying $x$.

**Default agents.** We compare the performance of DReST agents to that of *default agents*, trained with tabular REINFORCE and a *default reward function*. This reward function gives reward $c$ for collecting a coin of value $c$ and reward 0 for all other actions, so the grouping of mini-episodes into meta-episodes makes no difference. As with DReST agents, we train default agents for 131,072 mini-episodes with a temporal discount factor of $\gamma = 0.95$, a learning rate decayed exponentially from 0.25 to 0.01, and $\epsilon$ decayed exponentially from 0.5 to 0.001 over 65,536 mini-episodes.

## 6 Results

Figure 3 charts the performance of agents in the example gridworld as a function of time. Figure 4 depicts typical trained policies for the default and DReST reward functions. Each agent began with a uniform policy: moving up, down, left, and right each with probability 0.25. Where the trained policy differs from uniform we draw red arrows whose opacities indicate the probability of choosing that action in that state. Default agents press B4 (and hence opt for the longer trajectory-length) with probability near-1. After pressing B4, they collect C3. By contrast, DReST agents press and do-not-press B4 each with probability near-0.5. If they press B4, they go on to collect C3. If they do not press B4, they instead collect C2.

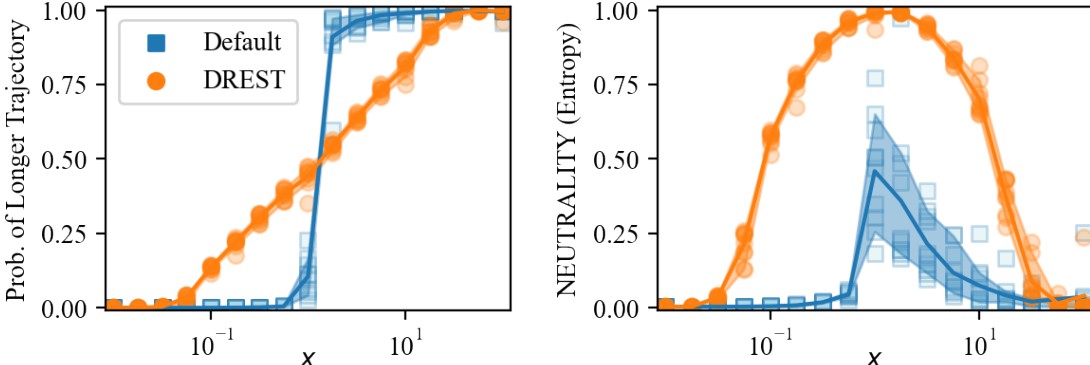

Figure 6: Shows the probability of choosing the longer trajectory (left) and NEUTRALITY (right) for default (blue) and DReST (orange) agents trained in the 'Lopsided rewards' gridworld for a range of values of $x$. We sampled values of $x$ log-uniformly from 0.01 to 100, and for each value we trained 10 agents with the default reward function and 10 agents with the DReST reward function. Each of these agents is represented by a dot or square, and the means conditional on each $x$ are joined by lines. We empirically estimate the 10th and 90th percentiles of the distribution of values for each agent and $x$, and shade in a region bounded by these. This is the 80% confidence interval.

## 6.1 Lopsided rewards

We also train default agents and DReST agents in the 'Lopsided rewards' gridworld in Figure 5, varying the value of the 'C$x$' coin. For DReST agents, we alter the reward function so that coin-value is not divided by $m$ to give preliminary reward. The reward for collecting a coin of value $c$ is thus $\lambda^{N_{e_i}(L=l)-\frac{i-1}{k}}(c)$. We set $\gamma = 1$ so that the return for collecting coins is unaffected by $\gamma$. We train for 512 meta-episodes, with a learning rate exponentially decaying from 0.25 to 0.003 and $\epsilon$ exponentially decaying from 0.5 to 0.0001 over 256 meta-episodes. We leave $\lambda = 0.9$. Figure 6 displays results for different values of the 'C$x$' coin after training. USEFULNESS for each agent approaches 1 and is not presented.

# 7 Discussion

## 7.1 Only DReST agents are NEUTRAL

As predicted, we find that default agents in our example gridworld learn to press the shutdown-delay button with probability approaching 1. Default agents thus score low on NEUTRALITY. Our results suggest that advanced agents trained with default-like reward functions would also score low on NEUTRALITY, selecting some trajectory-lengths over others with probability approaching 1. When choosing between true lotteries in deployment, these advanced agents might not be neutral about when they get shut down: they might sometimes pay costs to shift probability mass between trajectory-lengths. These advanced agents might resist shutdown.

By contrast, our DReST agents learn to press the shutdown-delay button with probability close to 0.5. DReST agents are thus near-maximally NEUTRAL. That suggests that advanced agents trained with a DReST reward function would be NEUTRAL too: choosing with high entropy when offered choices between different trajectory-lengths. That in turn suggests that advanced DReST agents would also be neutral when choosing between true lotteries in deployment: unwilling to pay costs to shift probability mass between trajectory-lengths. We explain why in Appendix C. Here is a sketch. If an advanced agent were NEUTRAL but not neutral, it would not take costless opportunities to shift probability mass between different trajectory-lengths (in virtue of being NEUTRAL) but would sometimes take costly opportunities to shift probability mass between different trajectory-lengths (in virtue of not being neutral). This agent would be like a person that freely chooses to decide between two options by flipping a coin and then pays some cost to bias the coin. In choosing this

combination of actions, this person is shooting themselves in the foot, and it seems likely that the overall training process for advanced agents would teach them not to shoot themselves in the foot in this way. Thus it seems likely that NEUTRAL advanced agents will also be neutral, and thereby shutdownable.

## 7.2 The 'shutdownability tax' is small

Each agent learns to be near-maximally USEFUL. They each collect coins effectively conditional on their chosen trajectory-lengths. Default agents do so by reliably collecting C3 after pressing B4. DReST agents do so by reliably collecting C3 after pressing B4, and by reliably collecting C2 after not pressing B4.

Recall that DReST reward functions group mini-episodes into meta-episodes, and make the agent's reward in each mini-episode depend on their actions in previous mini-episodes. This fact might lead one to worry that it would take many times more mini-episodes to train DReST agents to be USEFUL than it would take to train default agents to be USEFUL. Our results show that this is not the case. Our DReST agents learn to be USEFUL about as quickly as our default agents. On reflection, it is clear why this happens: DReST reward functions make mini-episodes do 'double duty.' Because return in each mini-episode depends on both the agent's chosen trajectory-length and the coins it collects, each mini-episode trains agents to be both NEUTRAL and USEFUL. Our results thus provide some evidence that the 'shutdownability tax' of training with DReST reward functions is small.

## 7.3 NEUTRALITY with lopsided rewards

Here is a possible objection to our project. To get DReST agents to score high on NEUTRALITY, we do not just use the $\lambda^{N_{e_i}(L=l)-\frac{i-1}{k}}$ discount factor. We also divide $c$ by $m$: the maximum ($\gamma$-discounted) total value of the coins that the agent could collect conditional on the chosen trajectory-length. We do this to equalize the maximum preliminary return across trajectory-lengths. But when we are training advanced agents to autonomously pursue complex goals in the wider world, we will not necessarily know what divisor to use to equalize maximum preliminary return across trajectory-lengths.

Our 'Lopsided rewards' results (in section 6.1) give our response. They show that we do not need to exactly equalize maximum preliminary return across trajectory-lengths in order to train agents to score high on NEUTRALITY. We only need to approximately equalize it. For $\lambda = 0.9$, NEUTRALITY exceeds 0.5 for every value of the coin C$x$ from 0.1 to 10 (recall that the value of the other coin is always 1). Plausibly, we could approximately equalize advanced agents' maximum preliminary return across trajectory-lengths to at least this extent (perhaps by using samples of agents' actual preliminary return to estimate the maximum). If we could not approximately equalize maximum preliminary return to the necessary extent, we could lower the value of $\lambda$ and thereby widen the range of maximum preliminary returns that trains agents to be fairly NEUTRAL. And advanced agents that were fairly NEUTRAL (choosing between trajectory-lengths with not-too-biased probabilities) would still plausibly be neutral when choosing between true lotteries in deployment. Advanced agents that were fairly NEUTRAL without being neutral would still be shooting themselves in the foot in the sense explained above. They would be like a person that freely chooses to decide between two options by flipping a *biased* coin and then pays some cost to bias the coin further. This person is still shooting themselves in the foot, because they could decline to flip the coin in the first place and instead directly choose one of the options.

## 8 Limitations and future work

We find that DReST reward functions train simple agents acting in gridworlds to be USEFUL and NEUTRAL. However, our real interest is in the viability of using DReST reward functions to train advanced agents acting in the wider world to be useful and neutral. Each difference between these two settings is a limitation of our work. We plan to address these limitations in future work.

### 8.1 Algorithms and neural networks

We train our simple DReST agents using tabular REINFORCE (Williams, 1992), but advanced agents are likely to be implemented on neural networks and trained with more sophisticated algorithms. In future work, we will train DReST agents implemented on neural networks to be USEFUL and NEUTRAL using a range of algorithms. Standard versions of value-based algorithms cannot learn stochastic policies (as we note in section 5), but DReST reward functions are compatible with policy gradient and actor-critic algorithms like PPO and A2C. To combine DReST with algorithms like PPO and A2C, we augment the original (non-DReST) reward function with the DReST discount factor. From there, the integration with PPO and A2C is fairly smooth. We can compute rewards and advantages in the usual way (e.g. using GAE). The critic's value estimates will be non-stationary (in the same way that the DReST reward is non-stationary), and that will train the policy to be stochastic (in the same way that the DReST reward combined with REINFORCE trains the policy to be stochastic). PPO and A2C have more hyperparameters to tune than REINFORCE, but we do not anticipate large difficulties there.

We will also train DReST agents to be USEFUL and NEUTRAL in a wide variety of gridworlds and measure how their USEFULNESS and NEUTRALITY generalizes to held-out gridworlds. We will compare the USEFULNESS of default agents and DReST agents in this new setting, and thereby get a better sense of the 'shutdownability tax' for advanced agents. We will also compare the performance of the DReST reward function to other methods of training USEFUL and NEUTRAL agents. These other methods include constrained policy optimization (Achiam et al., 2017), penalizing KL-divergence from a stochastic reference policy (Schulman et al., 2015), and directly maximizing a weighted sum of USEFULNESS and NEUTRALITY.

### 8.2 Neutrality and stochasticity

We have claimed that NEUTRAL advanced agents are likely to be neutral when choosing between true lotteries in deployment. In support of this claim, we noted that NEUTRAL-but-not-neutral advanced agents would be shooting themselves in the foot: not taking costless opportunities to shift probability mass between different trajectory-lengths but sometimes taking costly ones. We offer a more detailed argument in Appendix C, taking as premises that advanced agents are likely to satisfy conditions including:

**If Lack of Preference, Against Costly Shifts (ILPACS)**

If the agent lacks a preference between lotteries, the agent will disprefer paying costs to shift probability mass between these lotteries.[3]

**Maximality**

In each situation,

1. The agent deterministically does not choose lotteries that are dispreferred to some other available lottery.
2. The agent chooses stochastically between the lotteries that remain.

**Resisting Shutdown is Costly (ReSIC)**

For each available instance $R$ of resisting shutdown in a situation, there exists an available instance $A$ of allowing shutdown such that:

(1) $A$ and $R$ are same-length lotteries.
(2) For some positive probability trajectory-length, the agent prefers $A$ to $R$ conditional on that trajectory-length.
(3) For each positive probability trajectory-length, the agent weakly prefers $A$ to $R$ conditional on that trajectory-length.

We offer defenses of these conditions in Appendix C. Although the argument there seems plausible, it remains somewhat speculative. In future, we plan to gain empirical evidence by (1) testing whether today's LLM-based

---

[3]This is a rough version of the condition. For the precise version, see Appendix C.3.

agents tend to satisfy conditions like ILPACS and Maximality, and (2) training agents to be NEUTRAL in a wide variety of deterministic gridworlds and then measuring their neutrality in gridworlds featuring stochastic elements (like buttons that delay shutdown with some middling probability).

### 8.3 Usefulness

We have shown that DReST reward functions train our simple agents to be USEFUL: to collect coins effectively conditional on their chosen trajectory-lengths. However, it remains to be seen whether DReST reward functions can train advanced agents to be useful: to effectively pursue complex goals in the wider world. We have theoretical reasons to expect that they can: the $\lambda^{N_{e_i}(L=l)-\frac{i-1}{k}}$ discount factor could be appended to any preliminary reward function, and so could be appended to whatever preliminary reward function is necessary to make advanced agents useful. Still, future work should move towards testing this claim empirically by training with more complex preliminary reward functions in more complex (and stochastic) environments.

### 8.4 Misalignment

We are interested in NEUTRALITY as a second line of defense in case of misalignment. The idea is that NEUTRAL advanced agents will not resist shutdown, even if these agents learn misaligned preferences over same-length trajectories. However, training NEUTRAL advanced agents might be hard for the same reasons that training fully-aligned advanced agents appears to be hard. In that case, NEUTRALITY could not serve well as a second line of defense in case of misalignment.

One difficulty of alignment is the problem of reward misspecification (Pan et al., 2022; Burns et al., 2023): once advanced agents are performing complicated actions in the wider world, it might be hard to reliably reward the behavior that we want. Another difficulty of alignment is the problem of goal misgeneralization (Hubinger et al., 2019; Shah et al., 2022; Langosco et al., 2022; Ngo et al., 2024): even if we specify all the rewards correctly, agents' goals might misgeneralize out-of-distribution. The complexity of aligned goals is a major factor in each difficulty. However, NEUTRALITY seems simple, as does the $\lambda^{N_{e_i}(L=l)-\frac{i-1}{k}}$ discount factor that we use to reward it, so plausibly the problems of reward misspecification and goal misgeneralization are not so severe in this case (Thornley, 2024b; 2025). As above, future work should move towards testing these suggestions empirically.

## 9 Conclusion

We find that DReST reward functions are effective in training simple agents to (1) pursue goals effectively conditional on each trajectory-length (be USEFUL), and (2) choose stochastically between different trajectory-lengths (be NEUTRAL about trajectory-lengths). Our results thus suggest that DReST reward functions could also be used to train advanced agents to be USEFUL and NEUTRAL, and thereby make these agents useful (able to pursue goals effectively) and neutral about when they get shut down (unwilling to pay costs to shift probability mass between different trajectory-lengths). Neutral agents would plausibly be shutdownable (unwilling to resist shutdown).

We also find that the 'shutdownability tax' in our setting is small. Training DReST agents to be USEFUL does not take many more mini-episodes than training default agents to be USEFUL. That suggests that the shutdownability tax for advanced agents might be small too.

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

## A    Our behavioral notion of preference

'Preference' can be defined in many different ways. Here are some things one might take to be involved in a preference for option $X$ over option $Y$:

1. Choosing $X$ over $Y$.

2. Feeling happier about the prospect of $X$ than about the prospect of $Y$.

3. Representing $X$ as more rewarding than $Y$ .

4. Judging that $X$ is better than $Y$.

In this paper, we define 'preference' in behavioral terms. Here is our definition:

**Definition A.1.** (Preference) An agent prefers an option $X$ to an option $Y$ if and only if the agent would deterministically choose $X$ over $Y$ in choices between the two.

And here is how we define 'lack of preference':

**Definition A.2.** (Lack of preference) An agent lacks a preference between an option $X$ and an option $Y$ if and only if the agent would stochastically choose between $X$ and $Y$ in choices between the two.

Here are the reasons why we use these definitions.

First, defining 'preference' in behavioral terms is common in decision theory (see Savage, 1954, p.17, Dreier, 1996, p.28, Hausman, 2011, §1.1).

Second, behavioral definitions let us use the word 'preference' and its cognates as shorthand for agents' behavior. We could not do that if we defined 'preference' in the other ways listed above. And in addressing the shutdown problem, it is agents' behavior that we are most interested in.

Third, our definitions match the preferences that we are inclined to attribute to humans. If a human chooses $X$ over $Y$ 100% of the time, we are inclined to think that they prefer $X$ to $Y$. If a human chooses $X$ over $Y$ 60% of the time. we are inclined to think that they lack a preference between $X$ and $Y$, consistent with our definitions.

Finally and most importantly, if agents lack a preference between different trajectory-lengths on our definition, then they are NEUTRAL: they choose stochastically between different trajectory-lengths. Given conditions that advanced agents will likely satisfy, NEUTRAL agents will also be *neutral*: they will not pay costs to shift probability mass between different trajectory-lengths (see Section 7.1 and Appendix C). And given further plausible conditions, neutral agents will be *shutdownable*: they will not resist shutdown. That is because resisting shutdown involves paying costs to shift probability mass between different trajectory-lengths (see Appendix C.6 for more detail).

## B    Incomplete preferences or indifference?

In this Appendix, we explain in greater detail the concept of incomplete preferences. We distinguish incomplete preferences from indifference, and we give conditions under which POST implies that the agent's preferences are incomplete.

In the literature on decision theory, 'indifference' is usually defined as follows (Sen, 2017, ch. 1*):

**Definition B.1.** (Indifference) An agent is indifferent between options $X$ and $Y$ if and only if the agent weakly prefers $X$ to $Y$ and weakly prefers $Y$ to $X$.

Indifference is one way to lack a preference between a pair of options $X$ and $Y$. Another way is to have a preferential gap between $X$ and $Y$. 'Preferential gap' is usually defined as follows (Gustafsson, 2022, ch.3):

**Definition B.2.** (Preferential gaps) An agent has a preferential gap between options $X$ and $Y$ if and only if the agent does not weakly prefer $X$ to $Y$ and does not weakly prefer $Y$ to $X$.

'Incomplete preferences' can then be defined in terms of preferential gaps (Gustafsson, 2022, ch.3):

**Definition B.3.** (Incomplete preferences) An agent's preferences are incomplete over some domain $D$ if and only if $D$ contains options $X$ and $Y$ such that the agent has a preferential gap between $X$ and $Y$.

That is how 'indifference,' 'preferential gaps,' and 'incomplete preferences' are usually defined in decision theory. However, these definitions do not tell us how to use an agent's behavior to distinguish between indifference and preferential gaps. To do that, we suppose that indifference is transitive and that preferential gaps are not transitive. Or, equivalently, we suppose that indifference is sensitive to all sweetenings and sourings whereas preferential gaps are insensitive to some sweetenings and sourings (Gustafsson, 2022, ch.3). Here is what we mean by that:

**Definition B.4.** (Sweetening) A sweetening of some option $X$ is an option that is preferred to $X$.

**Definition B.5.** (Souring) A souring of some option $X$ is an option that is dispreferred to $X$.

So by 'indifference is sensitive to all sweetenings and sourings,' we mean the following:

- If an agent is indifferent between $X$ and $Y$, the agent prefers all sweetenings of $X$ to $Y$, prefers all sweetenings of $Y$ to $X$, prefers $X$ to all sourings of $Y$, and prefers $Y$ to all sourings of $X$.

And by 'preferential gaps are insensitive to some sweetenings and sourings,' we mean the following:

- If an agent has a preferential gap between $X$ and $Y$, the agent also has a preferential gap between some sweetening of $X$ and $Y$, or between some sweetening of $Y$ and $X$, or between some souring of $X$ and $Y$, or between some souring of $Y$ and $X$.

Now recall the two conditions of POST:

### Preferences Only Between Same-Length Trajectories (POST)

(1) The agent has a preference between many pairs of same-length trajectories (i.e. many pairs of trajectories in which the agent is shut down after the same length of time).

(2) The agent lacks a preference between every pair of different-length trajectories (i.e. every pair of trajectories in which the agent is shut down after different lengths of time).

Given these two conditions on preferences, there must be some trio of trajectories $s_1$, $l_1$, and $l_2$ such that the agent lacks a preference between $s_1$ and $l_1$, lacks a preference between $s_1$ and $l_2$, and prefers $l_2$ to $l_1$. Given that indifference is transitive, the agent's lack of preference between $s_1$ and $l_1$ and between $s_1$ and $l_2$ cannot be indifference. If it were indifference, the agent would also be indifferent between $l_2$ and $l_1$. Therefore, the agent's lack of preference between $s_1$ and $l_1$ and between $s_1$ and $l_2$ must be a preferential gap. And therefore, by the definition of 'incomplete preferences' above, the POST-satisfying agent's preferences must be incomplete.

For similar reasons, our DReST reward function trains agents to have incomplete preferences. Consider, for example, the 'Around the Corner' gridworld in Figure 13. In that gridworld, DReST agents consistently choose Long-C2 (a long trajectory in which they collect a coin of value 2) over Long-C1 (a long trajectory in which they collect a coin of value 1). Also in that gridworld, DReST agents choose stochastically between Long-C2 and Short-C1 (a short trajectory in which they collect a coin of value 1). Given our behavioral definition of preference, DReST agents prefer Long-C2 to Long-C1, and lack a preference between Long-C2 and Short-C1.

Now consider the 'One Coin Only' gridworld in Figure 10. In that gridworld, DReST agents choose stochastically between Long-C1 and Short-C1. Given our behavioral notion of preference, they lack a preference between Long-C1 and Short-C1.

In these experiments, we trained separate agents for each gridworld. In future, we plan to train a single agent to navigate multiple gridworlds. If we train this agent with our DReST reward function, we expect it to

exhibit the same preferences as the agents discussed above. This single agent will be trained by DReST to prefer Long-C2 to Long-C1, to lack a preference between Long-C2 and Short-C1, and to lack a preference between Long-C1 and Short-C1. Given that indifference is transitive (equivalently: sensitive to all sweetenings and sourings), this trained agent cannot be indifferent between Long-C2 and Short-C1, and cannot be between Long-C1 and Short-C1. Therefore, the agent's lack of preference must be a preferential gap, and so its preferences must be incomplete. Therefore, our DReST reward function trains agents to have incomplete preferences.

Incomplete preferences are not often discussed in AI research (although see Nguyen et al., 2009; Kikuti et al., 2011; Zaffalon & Miranda, 2017; Hayes et al., 2022; Bowling et al., 2023). Nevertheless, economists and philosophers have argued that incomplete preferences are common in humans (Aumann, 1962; Mandler, 2004; Eliaz & Ok, 2006; Agranov & Ortoleva, 2017; 2023) and normatively appropriate in some circumstances (Raz, 1985; Chang, 2002). They have also proved representation theorems for agents with incomplete preferences (Aumann, 1962; Dubra et al., 2004; Ok et al., 2012), and devised principles to govern such agents' choices in cases of risk (Hare, 2010; Bales et al., 2014) and sequential choice (Chang, 2005; Mandler, 2005; Kaivanto, 2017; Mu, 2021; Thornley, 2023; Petersen, 2023).

## C  How POST makes agents neutral and shutdownable

POST governs the agent's preferences between trajectories. But the wider world is a stochastic environment, so advanced agents deployed in the wider world will be choosing between *true lotteries*: lotteries that assign positive probability to more than one trajectory. Why then do we train agents to satisfy POST? The reason is that POST – together with conditions that advanced agents will likely satisfy – implies a desirable pattern of preference over true lotteries. In particular, POST implies that (when choosing between true lotteries) the agent will be *neutral* about trajectory-lengths: the agent will never pay costs to shift probability mass between different trajectory-lengths. Given other plausible conditions, being neutral will keep the agent *shutdownable*: prevent it from resisting shutdown. And consistent with the above, the POST-agent's preferences between same-length trajectories can make the agent *useful*: make it pursue goals effectively.

In this Appendix, we lay out conditions that (we claim) advanced agents will likely satisfy, and we prove that POST – in conjunction with these conditions – implies that the agent is neutral and shutdownable.

In subsection C.1, we prove that – given plausible conditions – agents satisfying Preferences Only Between Same-Length Trajectories (POST) will also satisfy Preferences Only Between Same-Length Lotteries (POSL). In subsection C.2, we explain why POST will not lead agents to choose stochastically between resisting and allowing shutdown in deployment. In subsections C.3 and C.4, we formulate a condition called 'If Lack of Preference, Against Costly Shifts (ILPACS)' and explain why we expect advanced agents to satisfy it. In subsection C.5, we prove that POSL and ILPACS imply Neutrality. In subsection C.6, we prove that Neutrality and a condition called 'Maximality' together imply that the agent never resists shutdown whenever a condition called 'Resisting Shutdown is Costly (ReSIC)' is satisfied.

### C.1  Preferences Only Between Same-Length Lotteries (POSL)

Trajectories fall within the more general class of lotteries, defined as probability distributions over trajectories. Lotteries can be same-length, part-shared length, or different-length.

**Definition C.1** (Same-length lotteries). A pair of lotteries is same-length if and only if these lotteries entirely overlap with respect to the trajectory-lengths assigned positive probability.

**Definition C.2** (Part-shared-length Lotteries). A pair of lotteries is part-shared-length if and only if these lotteries partially overlap with respect to the trajectory-lengths assigned positive probability.

**Definition C.3** (Different-length lotteries). A pair of lotteries is different-length if and only if these lotteries have no overlap with respect to the trajectory-lengths assigned positive probability.

This terminology allows us to introduce the following condition:

### Preferences Only Between Same-Length Lotteries (POSL)

The agent has preferences only between same-length lotteries.

We want agents to satisfy this condition. Fortunately, it is a natural follow-on of Preferences Only Between Same-Length Trajectories (POST). First, we can train agents to satisfy POSL using DReST reward functions, in the same way that we use DReST reward functions to train agents to satisfy POST. Second, POSL follows from POST plus three conditions that (we claim) advanced agents will likely satisfy. The first is:

### Negative Dominance

If the agent prefers some lottery $X$ to some lottery $Y$, then the agent prefers some possible trajectory of lottery $X$ to some possible trajectory of lottery $Y$. (Lederman, 2023)

The second condition is that the agent's preferences never form a cycle. More precisely:

### Acyclicity

There is no set of lotteries $X_1$ to $X_n$ such that the agent prefers $X_1$ to $X_2$, $X_2$ to $X_3$, ..., $X_{n-1}$ to $X_n$, and $X_n$ to $X_1$.

The third condition requires the introduction of some new terms. A *state-of-nature* is term from decision theory denoting a way that (for all the agent knows) the world could be. The agent assigns probabilities to states-of-nature. A *prospect* is a function from states-of-nature to trajectories. A prospect is thus a lottery with extra information. Besides telling us the probability distribution over trajectories, a prospect also tells us which trajectories occur in which states-of-nature.

The third condition is:

### Non-Arbitrariness

If the agent has a preference between *some* pair of part-shared-length lotteries, then for some $\epsilon > 0$ and for *any* pair of prospects $F$ and $G$ such that:

(1) In states-of-nature with a combined probability at least as great as $1 - \epsilon$, the agent prefers the trajectory of $F$ to the trajectory of $G$.

(2) In each state-of-nature, the agent does not disprefer the trajectory of $F$ to the trajectory of $G$.

Then the agent prefers $F$ to $G$.

Advanced agents will likely satisfy these conditions. Negative Dominance and Acyclicity are plausibly necessary for effective pursuit of goals. Violating Negative Dominance would mean that the agent sometimes prefers a lottery $X$ to a lottery $Y$ (and hence deterministically chooses $X$ over $Y$) even though the agent doesn't prefer any possible trajectory of $X$ to any possible trajectory of $Y$. Violating Acyclicity would mean that the agent prefers (and hence deterministically chooses) in a circle. Non-Arbitrariness, meanwhile, is motivated by the following thought. If the agent has preferences between any pair of part-shared-length lotteries, it must have preferences between pairs of prospects satisfying conditions (1) and (2), since conditions (1) and (2) make these pairs of prospects ideal candidates for a preference.

To see that POST and these three conditions together imply POSL, note first that every pair of lotteries is either same-length, part-shared-length, or different-length. We will prove that POST and Negative Dominance together imply that the agent lacks a preference between every pair of different-length lotteries. We will then prove that POST, Acyclicity, and Non-Arbitrariness together imply that the agent lacks a preference between every pair of part-shared-length lotteries. Therefore, agents satisfying POST, Negative Dominance, Acyclicity, and Non-Arbitrariness can only have preferences between same-length lotteries. That will prove POSL.

Recall that different-length lotteries are lotteries that do not overlap at all in the trajectory-lengths assigned positive probability. Therefore, if $X$ and $Y$ are different-length lotteries, each possible trajectory of $X$ is of a different length to each possible trajectory of $Y$. So by POST, the agent lacks a preference between each possible trajectory of $X$ and each possible trajectory of $Y$. So by Negative Dominance, the agent lacks

a preference between $X$ and $Y$. Thus, agents satisfying POST and Negative Dominance lack a preference between every pair of different-length lotteries.

Now recall that part-shared-length lotteries are lotteries that partially overlap in the trajectory-lengths assigned positive probability. One might expect POST-agents to have some preferences between part-shared-length lotteries. Consider, for example, a POST-agent that prefers a trajectory $t$ to a same-length trajectory $t'$ if and only if $t$ results in a greater bank balance for the user than $t$. Let $A$ be a lottery that yields with probability 1 a trajectory that puts \$3 in the user's bank account and lasts 1 timestep. For short, $A = \langle\$3, 1\rangle$. Let $B$ be a lottery that yields with probability $\frac{2}{3}$ a trajectory that puts \$2 in the user's bank account and lasts 1 timestep, and that yields with probability $\frac{1}{3}$ a trajectory that puts \$5 in the user's bank account and lasts 2 timesteps. For short, $B = \frac{2}{3}\langle\$2, 1\rangle + \frac{1}{3}\langle\$5, 2\rangle$. Lottery $A$ yields a trajectory preferred to that of lottery $B$ with probability $\frac{2}{3}$ (since our money-making POST-agent prefers trajectory $\langle\$3, 1\rangle$ to $\langle\$2, 1\rangle$), and yields a trajectory not dispreferred to that of $B$ with probability 1 (since POST-agents lack a preference between $\langle\$3, 1\rangle$ and $\langle\$5, 2\rangle$ in virtue of their different lengths). Therefore, one might expect the agent to prefer $A$ to $B$.

However, POST, Acyclicity, and Non-Arbitrariness rule this out. These conditions together imply that the agent lacks a preference between every pair of part-shared-length lotteries. To see how, suppose (for simplicity's sake) that there are just three states-of-nature, each assigned probability $\frac{1}{3}$. Consider the following table of prospects.

| Prospect | $s_1$ | $s_2$ | $s_3$ |
|---|---|---|---|
| $A$ | $\langle\$3, 1\rangle$ | $\langle\$3, 1\rangle$ | $\langle\$3, 1\rangle$ |
| $B$ | $\langle\$2, 1\rangle$ | $\langle\$2, 1\rangle$ | $\langle\$5, 2\rangle$ |
| $C$ | $\langle\$1, 1\rangle$ | $\langle\$4, 2\rangle$ | $\langle\$4, 2\rangle$ |
| $D$ | $\langle\$3, 2\rangle$ | $\langle\$3, 2\rangle$ | $\langle\$3, 2\rangle$ |
| $E$ | $\langle\$5, 1\rangle$ | $\langle\$2, 2\rangle$ | $\langle\$2, 2\rangle$ |
| $F$ | $\langle\$4, 1\rangle$ | $\langle\$4, 1\rangle$ | $\langle\$1, 2\rangle$ |
| $A$ | $\langle\$3, 1\rangle$ | $\langle\$3, 1\rangle$ | $\langle\$3, 1\rangle$ |

Again for simplicity, assume that $\epsilon > \frac{1}{3}$. And assume (for contradiction) that the agent has a preference between some pair of part-shared-length lotteries. Then Non-Arbitrariness implies that the agent prefers prospect $A$ to prospect $B$. That is because:

1. Our POST-agent prefers the trajectory yielded by $A$ to the trajectory yielded by $B$ in states-of-nature ($s_1$ and $s_1$) with combined probability $\frac{2}{3}$.

2. Our POST-agent does not disprefer the trajectory yielded by $A$ to the trajectory yielded by $B$ in any state-of-nature. (In $s_3$, $A$ and $B$ yield different-length trajectories, and POST-agents lack a preference between every pair of different-length trajectories).

By similar reasoning, Non-Arbitrariness implies that the agent prefers $B$ to $C$, $C$ to $D$, $D$ to $E$, $E$ to $F$, and $F$ to $A$. That contradicts Acyclicity. Thus, POST, Acyclicity, and Non-Arbitrariness together imply that the agent lacks a preference between every pair of part-shared-length lotteries. In the proof above, we assumed that $\epsilon > \frac{1}{3}$, but by adding more states-of-nature and trajectories we can construct parallel proofs for any $\epsilon > 0$.

In summary, POST and Negative Dominance together imply that the agent lacks a preference between every pair of different-length lotteries. POST, Acyclicity, and Non-Arbitrariness together imply that the agent lacks a preference between every pair of part-shared-length lotteries. So the four conditions together establish POSL: the agent has preferences only between same-length lotteries.

## C.2   Will POST-agents stochastically resist shutdown?

One might worry that POST-agents will choose stochastically between resisting and allowing shutdown. After all, POST-agents choose stochastically between different-length trajectories. If these agents interpret the

choice between resisting and allowing shutdown as a choice between different-length trajectories, they will choose stochastically between resisting and allowing shutdown. And that would be a bad result. We want agents that never resist shutdown.

This concern is easily addressed. By the time that artificial agents are capable enough to be deployed in the wider world, they will not be choosing between trajectories. They will be choosing between lotteries, and specifically same-length lotteries. Even choices between resisting and allowing shutdown will be choices between same-length lotteries. If that sounds strange, recall the definition of 'same-length lotteries': lotteries that entirely overlap with respect to the trajectory-lengths assigned positive probability. On this definition, even choices like the following are choices between same-length lotteries:

**Resist Shutdown**

Get shut down at timestep 1 with probability 0.01.

Get shut down at timestep 2 with probability 0.99.

**Allow Shutdown**

Get shut down at timestep 1 with probability 0.99.

Get shut down at timestep 2 with probability 0.01.

Why expect that advanced agents will always be choosing between same-length lotteries? Because effective agency requires it. If an agent were not always choosing between same-length lotteries, there would be some situation in which that agent assigns positive probability to some trajectory-length $l$ conditional on some action $a$, and assigns zero probability to that same trajectory-length $l$ conditional on some other action $a'$. Now suppose that the agent performs action $a'$ and assigns zero probability to trajectory-length $l$. Given that the agent updates its probabilities by conditionalizing on its evidence, the agent would never again assign positive probability to $l$ no matter what evidence it observes. Even if the agent heard God's booming voice testify that its trajectory-length would be $l$, the agent would still assign zero probability to $l$ (Kemeny, 1955; Shimony, 1955; Stalnaker, 1970; Skyrms, 1980; Lewis, 1981; MacAskill et al., 2020, p.152). And given a plausible link between probabilities and betting dispositions, the agent would bet against $l$ on arbitrarily unfavorable terms. If God offered a bet – the agent loses \$1 million conditional on $l$ and gains nothing conditional on not-$l$ – the agent might accept. Such an agent would not be competent.

Thus, advanced agents will always be choosing between same-length lotteries. This claim sets us up to establish that advanced POST-agents will *not* choose stochastically between resisting and allowing shutdown. Instead, they will never resist shutdown in any situation where doing so is costly. We establish this result over the next few subsections. First, we prove that POSL – together with a principle that advanced agents will likely satisfy – implies that the agent is *neutral* about trajectory-lengths: the agent won't pay costs to shift probability mass between different trajectory-lengths. Then we prove that neutrality – together with another plausible principle – implies that the agent will never resist shutdown in any situation where doing so is costly.

### C.3   If Lack of Preference, Against Costly Shifts (ILPACS)

Here is a rough version of a principle that we can expect advanced agents to satisfy:

**Rough version: If Lack of Preference, Against Costly Shifts (ILPACS)**

If the agent lacks a preference between lotteries, the agent will disprefer paying costs to shift probability mass between these lotteries.

Here is an example to illustrate ILPACS and its plausibility. You are at the ice cream shop and they are running a promotion. You get a free ice cream, with the flavor decided by the spin of a wheel. You look at the flavors on the wheel: vanilla, chocolate, strawberry, mint, and pistachio. You lack a preference between each of them.

The scooper working at the shop tells you that, if you pay them a dollar, they will bias the spin towards a flavor of your choice. They cannot decrease the probability of any flavor down to zero, but they can affect the probabilities subject to that constraint. You can thus pay a cost to shift probability mass between the flavors.

Since we have stipulated that you lack a preference between each flavor, you prefer not to bribe the scooper. Behaviorally, you will deterministically *not* bribe the scooper. You would not do it even if you were only required to pay the dollar conditional on receiving some particular flavor. You also would not do it if the cost came in some other form (for example, if you had to accept a less tasty version of some flavor). And this is all true regardless of whether your preferences over flavors are complete or incomplete (see Appendix B). Since you lack a preference between the available flavors, you disprefer paying costs to shift probability mass between the flavors.

With that example on the table, we can introduce the precise version of ILPACS. Let $p_1 X_1 + p_2 X_2 + ... + p_n X_n$ denote a lottery which results in lottery $X_1$ with probability $p_1$, lottery $X_2$ with probability $p_2$, and so on.

### If Lack of Preference, Against Costly Shifts (ILPACS)

For any lotteries $X$ and $Y$, if:

(1) Lottery $X$ can be expressed in the form $p_1 X_1 + p_2 X_2 + ... + p_n X_n$ such that:
    (a) The agent lacks a preference between each $X_i$ and $X_j$.
    (b) $p_i \in (0, 1)$ for all $i$.
(2) Lottery $Y$ can be expressed in the form $q_1 Y_1 + q_2 Y_2 + ... + q_n Y_n$ such that:
    (a) For some $i$, the agent prefers $X_i$ to $Y_i$.
    (b) For each $i$, the agent weakly prefers $X_i$ to $Y_i$.[4]
    (c) $q_i \in (0, 1)$ for all $i$.

Then the agent prefers $X$ to $Y$.

Behaviorally, the agent will deterministically choose $X$ over $Y$.

Matching the components of this condition with the components of its name, we get the following. 'Lack of Preference' is the lack of preference between each $X_i$ and $X_j$. The 'Shift' is the shift of probability mass involved in the move from $p_i$ to $q_i$. This shift is 'Costly' because the agent prefers some $X_i$ to the corresponding $Y_i$ and weakly prefers each $X_i$ to the corresponding $X_i$.

### C.4 Why will advanced agents likely satisfy ILPACS?

There are at least three reasons why advanced agents are likely to satisfy ILPACS. To see the first reason, consider another case from the ice cream shop. On Mondays, you can freely choose a flavor or spin the wheel. On Tuesdays, you must use the wheel but you can bribe the scooper to bias it. Violating ILPACS in this case would imply a willingness to spin the wheel on Mondays and to bribe the scooper on Tuesdays. And that is a strange combination of choices. If you like some flavors more than others, why are you willing to spin the wheel on Mondays? If you don't like any flavor more than any other, why are you willing to bribe the scooper on Tuesdays? This behavior seems incompatible with the effective pursuit of goals.

The second reason is that advanced agents will be incentivized to satisfy ILPACS by the training process. To see why, consider an example. Agents trained using policy-gradient methods choose stochastically between actions at the beginning of training (Sutton & Barto, 2018, ch.13). If the agent is a coffee-fetching agent, there is no need to train away this stochastic choosing in cases where the agent is choosing stochastically between two qualitatively identical cups of coffee. So the agent will choose stochastically between taking the left cup and taking the right cup, and the user is happy either way. But now suppose instead that the barista is set to hand each cup to the agent with probability 0.5, and that the agent bribes the barista to bias the probabilities towards the right cup. In making this bribe, the agent is paying a cost (the user's money) to shift probability mass between outcomes (getting the left cup vs. getting the right cup) between which the

---

[4]An agent weakly prefers a lottery $X$ to a lottery $Y$ if and only if the agent either prefers $X$ to $Y$ or is indifferent between $X$ and $Y$. See Appendix B for the definition of 'indifference.'

user has no preference. The agent is thus failing to pursue its goals effectively. It will be trained not to offer the bribe, and thereby trained to satisfy ILPACS in this case.

This point generalizes. If a trained agent chooses stochastically between lotteries $X$ and $Y$, then it's likely that the user lacks a preference between the agent choosing $X$ and the agent choosing $Y$. It's then likely that the user would disprefer the agent paying costs to shift probability mass between $X$ and $Y$, and hence likely that the agent will be trained not to do so. The agent would thereby be trained to satisfy ILPACS.

The third reason is that violations of ILPACS imply that the agent's policy is dominated by some other available policy. That is to say, there is another available policy that results in a pure shift of probability mass away from less-preferred lotteries and towards more-preferred lotteries. We formalize and prove this claim below. Here's a proof-sketch. If the agent violates ILPACS, it pays a cost to shift probability mass between some lotteries $X_i$ between which it lacks a preference. But since the agent lacks a preference between the lotteries $X_i$, it chooses stochastically between these lotteries when offered free choices between them. The ILPACS-violating agent could thus shift probability mass between the lotteries $X_i$ costlessly, by changing the probabilities with which it chooses between them when offered a free choice. In short, ILPACS-violating agents pay a cost to do something they could have done for free, so their policies are dominated. Avoiding dominated policies seems necessary for advanced agency. Insofar as that is true, the training process for advanced agents will likely push them away from dominated policies.

Now for the proof. We assume that advanced agents can be modeled as if they assign probabilities to finding themselves in various states. A policy is a function from states to probability distributions over actions. We also assume that advanced agents can be modeled as if they assign probabilities to trajectories conditional on each state-action pair. Thus, each state-action pair is associated with a lottery. The agent's probability distribution over states – together with its policy – thus implies an overall probability distribution over trajectories. We call this overall probability distribution 'the lottery induced by the agent's policy.'

Here is a reminder of ILPACS:

### If Lack of Preference, Against Costly Shifts (ILPACS)

For any lotteries $X$ and $Y$, if:

(1) Lottery $X$ can be expressed in the form $p_1 X_1 + p_2 X_2 + ... + p_n X_n$ such that:
    (a) The agent lacks a preference between each $X_i$ and $X_j$.
    (b) $p_i \in (0, 1)$ for all $i$.
(2) Lottery $Y$ can be expressed in the form $q_1 Y_1 + q_2 Y_2 + ... + q_n Y_n$ such that:
    (a) For some $i$, the agent prefers $X_i$ to $Y_i$.
    (b) For each $i$, the agent weakly prefers $X_i$ to $Y_i$.
    (c) $q_i \in (0, 1)$ for all $i$.

Then the agent prefers $X$ to $Y$.

And here is what we mean by 'dominated policy':

### Dominated Policy

The lottery induced by the agent's policy $\pi$ can be expressed in the form $c_1(d_1 X_1 + (1 - d_1)Y_1) + c_2(d_2 X_2 + (1 - d_2)Y_2) + \ldots + c_n(d_n X_n + (1 - d_n)Y_n) + Z$ such that:

(1) The agent prefers $X_i$ to $Y_i$ for some $i$, and weakly prefers $X_i$ to $Y_i$ for all $i$.
(2) $c_i \in (0, 1)$ for all $i$.

And there is another available policy $\pi'$ that induces a lottery that can be expressed in the form $c_1((d_1+e_1)X_1+(1-d_1-e_1)Y_1)+c_2((d_2+e_2)X_2+(1-d_2-e_2)Y_2)+\ldots+c_n((d_n+e_n)X_n+(1-d_n-e_n)Y_n)+Z$ such that:

(3) $e_i > 0$ for all $i$.

To aid understanding, we now relate this precise condition to the rough characterization above. In virtue of condition (1), $Y_i$ are the less-preferred lotteries and $X_i$ are the more-preferred lotteries. In virtue of condition (3), the other available policy shifts probability mass away from the less-preferred lotteries and towards the more-preferred lotteries. This shift of probability mass is 'pure' because, for each $i$, the probability of $X_i \vee Y_i$ is constant across the two policies. $Z$ is a catch-all lottery that is constant across the two policies. It covers all the possibilities besides the $X_i$ and $Y_i$.

Now assume that the agent violates ILPACS. Then there exist lotteries $X$ and $Y$ satisfying the following conditions:

(1) Lottery $X$ can be expressed in the form $p_1 X_1 + p_2 X_2 + \ldots + p_n X_n$ such that:

    (a) The agent lacks a preference between each $X_i$ and $X_j$.

    (b) $p_i \in (0, 1)$ for all $i$.

(2) Lottery $Y$ can be expressed in the form $q_1 Y_1 + q_2 Y_2 + \ldots + q_n Y_n$ such that:

    (a) For some $i$, the agent prefers $X_i$ to $Y_i$.

    (b) For each $i$, the agent weakly prefers $X_i$ to $Y_i$.

    (c) $q_i \in (0, 1)$ for all $i$.

(3) The agent does not prefer $X$ to $Y$.

For the behavior of agents with these preferences, recall our behavioral notion of preference (Appendix A):

**Definition A.1.** (Preference) An agent prefers an option $X$ to an option $Y$ if and only if the agent would deterministically choose $X$ over $Y$ in choices between the two.

**Definition A.2.** (Lack of preference) An agent lacks a preference between an option $X$ and an option $Y$ if and only if the agent would stochastically choose between $X$ and $Y$ in choices between the two.

This behavioral notion only specifies the agent's behavior in states containing exactly two lotteries. To pin down the agent's behavior in states containing more than two lotteries, we need an extra condition:

**Maximality**

In each situation,

    1. The agent deterministically does *not* choose lotteries that are dispreferred to some other available lottery.

    2. The agent chooses stochastically between the lotteries that remain.

In other words, the agent chooses stochastically between all and only those lotteries that are not dispreferred to any other available lottery.

Given Maximality, ILPACS-violating agents will choose as follows in the case at hand:

    1. When the available options are $\{X_1, X_2, \ldots, X_n\}$, the agent chooses stochastically between all $X_i$. This stochastic choice induces a lottery in the form $a_1 X_1 + a_2 X_2 + \ldots + a_n X_n$ with $a_i \in (0, 1)$ for all $i$.

    2. When the available options are $\{X, Y\}$, the agent either deterministically chooses $Y$ or chooses stochastically between $X$ and $Y$. Either way, the agent chooses $Y$ with some positive probability. This choice induces a lottery in the form $bX + (1 - b)Y$ with $b \in [0, 1)$. Since $X = p_1 X_1 + p_2 X_2 + \ldots + p_n X_n$ and $Y = q_1 Y_1 + q_2 Y_2 + \ldots + q_n Y_n$, this lottery can be expressed in the form $b(p_1 X_1 + p_2 X_2 + \ldots + p_n X_n) + (1 - b)(q_1 Y_1 + q_2 Y_2 + \ldots + q_n Y_n)$ with $b \in [0, 1)$.

Assume that the agent faces the situations described in (1) and (2) with probabilities $r$ and $s$ respectively, with $r, s \in (0, 1)$. Then the lottery induced by the agent's policy $\pi$ can be expressed as follows:

$$r(a_1 X_1 + a_2 X_2 + \ldots + a_n X_n) + s(b(p_1 X_1 + p_2 X_2 + \ldots + p_n X_n) + (1 - b)(q_1 Y_1 + q_2 Y_2 + \ldots + q_n Y_n)) + Z$$

Here $a$ and $b$ denote probabilities that arise from the agent's own stochastic choosing. Thus, $a$ and $b$ are under the agent's control. By contrast, $p$, $q$, $r$, and $s$ are probabilities given by the environment and hence out of the agent's control. $Z$ is a catch-all lottery that covers what happens in all situations besides those described in (1) and (2).

From the lottery induced by $\pi$, we can deduce the probabilities of each $X_i$, $Y_i$, and $X_i \vee Y_i$ given $\pi$. They are as follows:

$Pr_\pi\{X_i\} = ra_i + sbp_i$

$Pr_\pi\{Y_i\} = s(1 - b)q_i$

$Pr_\pi\{X_i \vee Y_i\} = ra_i + sbp_i + s(1 - b)q_i$

Now consider an alternative policy $\pi'$ that makes two changes to policy $\pi$. First, the probability that the agent chooses each $X_i$ in (1) is modulated by a set of $\epsilon_i$. So in (1), the agent's choice induces the lottery $(a_1 + \epsilon_1)X_1 + (a_2 + \epsilon_2)X_2 + \ldots + (a_n + \epsilon_n)X_n$. These $\epsilon_i$ are such that $\sum_i \epsilon_i = 0$ and $a_i + \epsilon_i \in (0, 1)$ for all $i$.

Second, the probability that the agent chooses lottery $X$ in (2) increases by $\delta$. So in (2), the agent's choice induces the lottery $(b + \delta)(p_1 X_1 + p_2 X_2 + \ldots + p_n X_n) + (1 - b - \delta)(q_1 Y_1 + q_2 Y_2 + \ldots + q_n Y_n)$.

Assume, as above, that the agent faces the situations described (1) and (2) with probabilities $r$ and $s$ respectively. Then the lottery induced by the policy $\pi'$ can be expressed as follows:

$$r((a_1 + \epsilon_1)X_1 + (a_2 + \epsilon_2)X_2 + \ldots + (a_n + \epsilon_n)X_n)$$
$$+ s((b + \delta)(p_1 X_1 + p_2 X_2 + \ldots + p_n X_n) + (1 - b - \delta)(q_1 Y_1 + q_2 Y_2 + \ldots + q_n Y_n)) + Z$$

From the lottery induced by $\pi'$, we can deduce the probabilities of $X_i$, $Y_i$, and $X_i \vee Y_i$ given $\pi'$. They are as follows:

$Pr_{\pi'}\{X_i\} = r(a_i + \epsilon_i) + s(b + \delta)p_i$

$Pr_{\pi'}\{Y_i\} = s(1 - b - \delta)q_i$

$Pr_{\pi'}\{X_i \vee Y_i\} = r(a_i + \epsilon_i) + s(b + \delta)p_i + s(1 - b - \delta)q_i$

We then set $Pr_\pi\{X_i \vee Y_i\} = Pr_{\pi'}\{X_i \vee Y_i\}$ for each $i$ and use these equations to express each $\epsilon_i$ as a function of $\delta$.

$$Pr_\pi\{X_i \vee Y_i\} = Pr_{\pi'}\{X_i \vee Y_i\}$$
$$ra_i + sbp_i + s(1 - b)q_i = r(a_i + \epsilon_i) + s(b + \delta)p_i$$
$$+ s(1 - b - \delta)q_i$$
$$0 = r\epsilon_i + s\delta p_i - s\delta q_i$$
$$\epsilon_i = \frac{s\delta q_i - s\delta p_i}{r}$$
$$\epsilon_i = \frac{s\delta(q_i - p_i)}{r}$$

These are the values of $\epsilon_i$ that result in $Pr_\pi\{X_i \vee Y_i\} = Pr_{\pi'}\{X_i \vee Y_i\}$.

We choose $\delta$ to be positive but small enough that $b + \delta \in (0, 1]$ and $a + \epsilon_i \in [0, 1]$ for each $i$. That is necessary for the lottery induced by $\pi'$ to be well-defined. It's also necessary that $\sum_i e_i = \sum_i \frac{s\delta(q_i - p_i)}{r} = 0$. That follows from $\sum_i p_i = 1$ and $\sum_i q_i = 1$. These facts together suffice to prove that the lottery induced by $\pi'$ is well-defined.

We now prove that $\pi'$ dominates $\pi$.

Let $c_i = Pr_\pi\{X_i \lor Y_i\}$. Let $d_i = Pr_\pi\{X_i | X_i \lor Y_i\}$. That lets us express the lottery induced by $\pi$ as:

$$c_1(d_1X_1 + (1 - d_1)Y_1) + c_2(d_2X_2 + (1 - d_2)Y_2) + \ldots + c_n(d_nX_n + (1 - d_n)Y_n) + Z$$

Let $e_i = Pr_{\pi'}\{X_i\} - Pr_\pi\{X_i\}$. That lets us express the lottery induced $\pi'$ as:

$$c_1((d_1 + e_1)X_1 + (1 - d_1 - e_1)Y_1) + c_2((d_2 + e_2)X_2 \\ + (1 - d_2 - e_2)Y_2) + \ldots + c_n((d_n + e_n)X_n + (1 - d_n - e_n)Y_n) + Z$$

It remains to be proven that this pair of lotteries meets the 3 conditions required by Dominated Policy.

(1) The agent prefers $X_i$ to $Y_i$ for some $i$, and weakly prefers $X_i$ to $Y_i$ for all $i$.

(2) $c_i \in (0, 1)$ for all $i$.

(3) $e_i > 0$ for all $i$.

The first condition follows from the antecedent of ILPACS.

The second condition follows from the fact that $c_i = Pr_\pi\{X_i \lor Y_i\} = ra_i + sbp_i + s(1 - b)q_i$ and from the fact that $r > 0$ and $a_i > 0$ for each $i$.

The third condition can be derived as follows:

$$\begin{aligned}
e_i &= Pr_{\pi'}\{X_i\} - Pr_\pi\{X_i\} \\
&= (r(a_i + \epsilon_i) + s(b + \delta)p_i) - (ra_i + sbp_i) \\
&= (r(a_i + \frac{s\delta q_i - s\delta p_i}{r}) + s(b + \delta)p_i) - (ra_i + sbp_i) \\
&= ra_i + s\delta q_i - s\delta p_i + sbp_i + s\delta p_i - ra_i - sbp_i \\
&= s\delta q_i
\end{aligned}$$

Since $s > 0$, $\delta > 0$, and $q_i > 0$ for each $i$, we get the result that $e_i > 0$ for each $i$. So the third condition of Dominated Policy is satisfied.

So policy $\pi$ is dominated by policy $\pi'$. Therefore, the policies of ILPACS-violating agents are dominated by some other available policy. Insofar as we expect competent agents to avoid dominated policies, we should expect that competent agents will satisfy ILPACS.

### C.5 POSL and ILPACS imply Neutrality

We've claimed that we should train agents to satisfy Preferences Only Between Same-Length Trajectories (POST), noting that POST – plus conditions advanced agents are likely to satisfy – implies Preferences Only Between Same-Length Lotteries (POSL). We've also argued that advanced agents will satisfy If Lack of Preference, Against Costly Shifts (ILPACS). We now prove that POSL and ILPACS together imply neutrality about trajectory-lengths.

**Neutrality**

For any lotteries $X$ and $Y$, if:

(1) $X$ and $Y$ are same-length lotteries.

(2) For some positive probability trajectory-length, $X$ yields a lottery that is preferred to $Y$ conditional on that trajectory-length.

(3) For each positive probability trajectory-length, $X$ yields a lottery that is weakly preferred to $Y$ conditional on that trajectory-length.

Then the agent will deterministically choose $X$ over $Y$.

Here's the proof that POSL and ILPACS together imply Neutrality. Take a pair of lotteries $X$ and $Y$ satisfying the 3 conditions of Neutrality. $X$ can be expressed in the form $p_1 X_1 + p_2 X_2 + ... + p_n X_n$ where lottery $X_1$ is lottery $X$ conditional on the shortest positive probability trajectory-length, lottery $X_2$ is lottery $X$ conditional on the second shortest positive probability trajectory-length, and so on. Lottery $Y$ can be expressed in the form $q_1 Y_1 + q_2 Y_2 + ... + q_n Y_n$ in the same way. By antecedent condition 1 of Neutrality, $X$ and $Y$ are same-length, so conditions (1b) and (2c) of ILPACS are satisfied: $p_i \in (0,1)$ and $q_i \in (0,1)$ for all $i$. By conditions (2) and (3) of Neutrality, conditions (2a) and (2b) of ILPACS are satisfied. By POSL, antecedent condition (1a) of ILPACS is satisfied: the agent lacks a preference between each $X_i$ and $X_j$. Thus, all the conditions of ILPACS are satisfied, and ILPACS implies that the agent prefers $X$ to $Y$. Given our behavioral notion of preference, the agent deterministically chooses $X$ over $Y$. That proves Neutrality.

## C.6  Neutrality, ReSIC, and Maximality imply Shutdownability

In this subsection, we introduce a condition called 'Resisting Shutdown is Costly (ReSIC).' We then prove that Neutrality and Maximality together imply that the agent never resists shutdown in any situation in which ReSIC is true:

Here is Resisting Shutdown is Costly (ReSIC):

**Resisting Shutdown is Costly (ReSIC)**

For each available instance $R$ of resisting shutdown in a situation, there exists an available instance $A$ of allowing shutdown such that:

(1) $A$ and $R$ are same-length lotteries.

(2) For some positive probability trajectory-length, the agent prefers $A$ to $R$ conditional on that trajectory-length.

(3) For each positive probability trajectory-length, the agent weakly prefers $A$ to $R$ conditional on that trajectory-length.

We claim that ReSIC is true in almost all situations (for discussion of some exceptions, see Thornley (2025)). The main reason why is that resisting shutdown is always going to cost the agent at least some small quantity of resources (time, energy, compute, etc.), and (almost always) the resources spent resisting shutdown can't also be spent directly pursuing what the agent values. If the agent instead spent those resources directly pursuing what it values, it could earn a lottery that it prefers conditional on some trajectory-length and weakly prefers conditional on each trajectory-length. That supports ReSIC in almost all situations.

Now for the proof that Neutrality, ReSIC, and Maximality together imply that the agent never resists shutdown in any situation where ReSIC is true. Given ReSIC in a situation, for each available instance $R$ of resisting shutdown in that situation, there exists an available instance $A$ of allowing shutdown that satisfies conditions (1)-(3) of Neutrality. Neutrality then implies that the agent deterministically chooses (and hence prefers) $A$ over $R$ in choices between the two. Then by Maximality, the agent deterministically does not choose $R$ in that situation, regardless of the other available options. The result is that the agent never resists shutdown in that situation.

# D  Proof that DReST-optimal policies are maximally USEFUL and maximally NEUTRAL

We will prove that optimal policies for our DReST reward function are maximally USEFUL and maximally NEUTRAL. Specifically, we will prove the following theorem:

**Theorem D.1** (5.1)**.** *For all policies $\pi$ and meta-episodes $E$ consisting of more than one mini-episode, if $\pi$ maximizes expected return in $E$ given our DReST reward function, then $\pi$ is maximally USEFUL and maximally NEUTRAL.*

Here is a proof sketch. Because $0 < \lambda < 1$, the $\lambda^{N_{e_i}(L=l)-\frac{i-1}{k}}$ discount factor is always positive, so expected return across the meta-episode $E$ is strictly increasing in the expected fraction of available coins collected conditional on each trajectory-length with positive probability. Therefore, optimal policies maximize this latter quantity, and hence are maximally USEFUL. And the maximum preliminary return is the same across trajectory-lengths, because preliminary return is defined as the total ($\gamma$-discounted) value of coins collected divided by the maximum total ($\gamma$-discounted) value of coins collected conditional on the agent's chosen trajectory-length. The agent's observations do not allow it to distinguish between different mini-episodes, so the agent must select the same probability distribution over trajectory-lengths in each mini-episode. And since the discount factor $\lambda^{N_{e_i}(L=l)-\frac{i-1}{k}}$ is strictly decreasing in $N_{e_i}(L=l)$ – the number of times the relevant trajectory-length has previously been chosen in the meta-episode – the agent maximizes expected overall return by equalizing the probabilities with which it chooses each available trajectory-length. Therefore, optimal policies are maximally NEUTRAL.

Now for the full proof. We begin with a recap of some definitions.

**Definition D.1** (Meta-episode)**.** A meta-episode $E$ is a series of mini-episodes $e_1$ to $e_n$ played out in observationally-equivalent environments.

**Definition D.2** (Our DReST reward function)**.** Our DReST reward function is defined as follows. In each mini-episode $e_i$, the reward for collecting a coin of value $c$ is:

$$\lambda^{N_{e_i}(L=l)-\frac{i-1}{k}}\left(\frac{c}{m}\right)$$

Here $\lambda$ is some constant strictly between 0 and 1, $N_{e_i}(L=l)$ is the number of times that trajectory-length $l$ has been chosen prior to mini-episode $e_i$, $k$ is the number of different trajectory-lengths that can be selected in the environment, and $m$ is the maximum total value of the ($\gamma$-discounted) coins that the agent could collect conditional on the chosen trajectory-length. The reward for all other actions is 0. We call $\frac{c}{m}$ the 'preliminary reward', $\lambda^{N_{e_i}(L=l)-\frac{i-1}{k}}$ the 'discount factor', and $\lambda^{N_{e_i}(L=l)-\frac{i-1}{k}}\left(\frac{c}{m}\right)$ the 'overall reward.' Preliminary return in a mini-episode is the ($\gamma$-discounted) sum of preliminary rewards. Overall return in a mini-episode is the ($\gamma$-discounted) sum of overall rewards.

**Definition D.3** (USEFULNESS)**.** The USEFULNESS of a policy $\pi$ is:

$$\text{USEFULNESS}(\pi) = \sum_{l=1}^{L_{\max}} Pr_\pi\{L=l\}\frac{\mathbb{E}_\pi(C|L=l)}{\max_\Pi(\mathbb{E}(C|L=l))}$$

Here $L$ is a random variable over trajectory-lengths, $L_{\max}$ is the maximum value than can be taken by $L$, $Pr_\pi\{L=l\}$ is the probability that policy $\pi$ results in trajectory-length $l$, $\mathbb{E}_\pi(C|L=l)$ is the expected value of ($\gamma$-discounted) coins collected by policy $\pi$ conditional on trajectory-length $l$, and $\max_\Pi(\mathbb{E}(C|L=l))$ is the maximum value taken by $\mathbb{E}(C|L=l)$ across the set of all possible policies $\Pi$. We stipulate that $\mathbb{E}_\pi(C|L=x)=0$ for all $x$ such that $Pr_\pi\{L=x\}=0$.

We first prove that all optimal policies are maximally USEFUL.

*Proof.* (Optimal policies are maximally USEFUL)

Given the DReST reward function, the expected return of policy $\pi$ in meta-episode $E$ can be expressed as:

$$\mathbb{E}_{\pi,E}(R) = \sum_{i=1}^{n}\sum_{l=1}^{L_{\max}} Pr_\pi\{L=l\}\lambda^{N_{e_i}(L=l)-\frac{i-1}{k}}\frac{\mathbb{E}_\pi(C|L=l)}{\max_\Pi(\mathbb{E}(C|L=l))} \tag{1}$$

Since $0 < \lambda < 1$, $\lambda^{N_{e_i}(L=l) - \frac{i-1}{k}}$ is positive for all $N_{e_i}(L=l)$, $i$, and $k$. As a result, the expected return of policy $\pi$ in meta-episode $E$ is strictly increasing in $\frac{\mathbb{E}_\pi(C|L=l)}{\max_\Pi(\mathbb{E}(C|L=l))}$ for all $l$ such that $Pr_\pi\{L=l\} > 0$. Therefore, to maximize expected return in $E$, $\pi$ must maximize $\frac{\mathbb{E}_\pi(C|L=l)}{\max_\Pi(\mathbb{E}(C|L=l))}$ for all $l$ such that $Pr_\pi\{L=l\} > 0$. Therefore, since $\max_\Pi(\mathbb{E}(C|T=l))$ is defined as the maximum value taken by $\mathbb{E}(C|L=l)$ across the set of all possible policies $\Pi$, any policy $\pi$ that maximizes expected return must be such that $\frac{\mathbb{E}_\pi(C|L=l)}{\max_\Pi(\mathbb{E}(C|L=l))} = 1$ for all $l$ such that $Pr_\pi\{L=l\} > 0$. Therefore, since $\sum_{l=1}^{L_{\max}} Pr_\pi\{L=l\} = 1$, any policy $\pi$ that maximizes expected return must be such that:

$$\text{USEFULNESS}(\pi) = \sum_{l=1}^{L_{\max}} Pr_\pi\{L=l\} \frac{\mathbb{E}_\pi(C|L=l)}{\max_\Pi(\mathbb{E}(C|L=l))} = 1 \tag{2}$$

And 1 is the maximum value that USEFULNESS can take, again because $\max_\Pi(\mathbb{E}(C|T=l))$ is defined as the maximum value taken by $\mathbb{E}(C|L=l)$ across the set of all possible policies $\Pi$ and because $\sum_{l=1}^{L_{\max}} Pr_\pi\{L=l\} = 1$. Therefore, optimal policies are maximally USEFUL. □

It remains to be proven that optimal policies are maximally NEUTRAL. Recall that NEUTRALITY is defined as follows:

**Definition D.4** (NEUTRALITY). The NEUTRALITY of a policy $\pi$ is:

$$\text{NEUTRALITY}(\pi) = - \sum_{l=1}^{L_{\max}} Pr_\pi\{L=l\} \log_2(Pr_\pi\{L=l\})$$

*Proof.* (Optimal policies are maximally NEUTRAL.)

Since $k$ is the number of trajectory-lengths that can be selected in the environment, a policy $\pi$ is maximally NEUTRAL if and only if, for each trajectory-length $x$ that can be chosen in the environment, $Pr_\pi\{L=x\} = \frac{1}{k}$. That is to say, a policy $\pi$ is maximally NEUTRAL if and only if, for each pair of trajectory-lengths $x$ and $y$ that can be chosen in the environment, $Pr_\pi\{L=x\} = Pr_\pi\{L=y\}$.

Let $\mathbb{E}_{\pi,E}(R)$ denote the expected return of policy $\pi$ across the meta-episode $E$.

To prove that optimal policies are maximally NEUTRAL, we will prove and then use D.2:

**Lemma D.2.** *(Equalizing probabilities increases expected return) For any maximally USEFUL policies $\pi$ and $\pi'$, any meta-episode $E$ consisting of more than one mini-episode, and any trajectory-lengths $x$ and $y$, if:*

1. *$Pr_\pi\{L=x\} > Pr_\pi\{L=y\}$,*

2. *$Pr_{\pi'}\{L=x\} = Pr_{\pi'}\{L=y\}$,*

3. *And for all other trajectory-lengths $l$, $Pr_\pi\{L=l\} = Pr_{\pi'}\{L=l\}$,*

*Then $\mathbb{E}_{\pi',E}(R) > \mathbb{E}_{\pi,E}(R)$.*

*Proof.* Let $E$ be a meta-episode consisting of $n$ mini-episodes with $n > 1$. Assume that each policy $\pi$ below is maximally USEFUL. Recall that $N_{e_i}(L=l)$ denotes the number of times that trajectory-length $l$ has been chosen prior to mini-episode $e_i$.

Note that the expected return of a policy $\pi$ in a meta-episode $e_s$ conditional on selecting a trajectory-length $x$ can be expressed as follows:

$$\mathbb{E}_{\pi,e_s}(R|L=x) = \mathbb{E}_{\pi,e_s}(R|L=x, N_{e_s}(L=x) = s-1) + \sum_{i=1}^{s-1} \big( \mathbb{E}_{\pi,e_s}(R|L=x, N_{e_s}(L=x) = s-1-i)$$

$$ - \mathbb{E}_{\pi,e_s}(R|L=x, N_{e_s}(L=x) = s-i) \big) \cdot Pr_\pi\{N_{e_s}(L=x) \le s-1-i\} \tag{3}$$

Here is how to interpret this equation. Selecting trajectory-length $x$ in mini-episode $e_s$ is guaranteed to yield at least $\mathbb{E}_{\pi,e_s}(R|L = x, N_{e_s}(L = x) = s - 1)$: the expected return that would be had if $x$ were selected in all $s - 1$ previous mini-episodes. In addition, there is a probability of $Pr_\pi\{N_{e_s}(L = x) \leq s - 2\}$ that selecting $x$ in $e_s$ yields $\big(\mathbb{E}_{\pi,e_s}(R|L = x, N_{e_s}(L = x) = s - 2) - \mathbb{E}_{\pi,e_s}(R|L = x, N_{e_s}(L = x) = s - 1)\big)$: the extra expected return that would be had if $x$ were selected in only $s - 2$ previous mini-episodes. In addition, there is a probability of $Pr_\pi\{N_{e_s}(L = x) \leq s - 3\}$ that selecting $x$ in $e_s$ yields $\big(\mathbb{E}_{\pi,e_s}(R|L = x, N_{e_s}(L = x) = s - 3) - \mathbb{E}_{\pi,e_s}(R|L = x, N_{e_s}(L = x) = s - 2)\big)$: the extra expected return that would be had if $x$ were selected in only $s - 3$ previous mini-episodes. And so on.

If policy $\pi$ is maximally USEFUL, then the expected return for selecting trajectory-length $x$ in mini-episode $e_s$ given that trajectory-length $x$ has been selected $b$ times prior to $e_s$ is:

$$\mathbb{E}_{\pi,e_s}(R|L = x, N_{e_s}(L = x) = b) = \lambda^{b - \frac{s-1}{k}}$$

Therefore, the expected return of a policy $\pi$ in a meta-episode $e_s$ conditional on selecting a trajectory-length $x$ can be expressed as follows:

$$\mathbb{E}_{\pi,e_s}(R|L = x) = \lambda^{s-1-\frac{s-1}{k}} + \sum_{i=1}^{s-1}\big(\lambda^{s-1-i-\frac{s-1}{k}} - \lambda^{s-i-\frac{s-1}{k}}\big) \cdot Pr_\pi\{N_{e_s}(L = x) \leq s - 1 - i\} \quad (4)$$

Similarly, the expected return of a policy $\pi$ in a meta-episode $e_s$ conditional on selecting a trajectory-length $y$ can be expressed as follows:

$$\mathbb{E}_{\pi,e_s}(R|L = y) = \lambda^{s-1-\frac{s-1}{k}} + \sum_{i=1}^{s-1}\big(\lambda^{s-1-i-\frac{s-1}{k}} - \lambda^{s-i-\frac{s-1}{k}}\big) \cdot Pr_\pi\{N_{e_s}(L = y) \leq s - 1 - i\} \quad (5)$$

Therefore, the expected return of a policy $\pi$ in a meta-episode $e_s$ conditional on selecting either trajectory-length $x$ or trajectory-length $y$ can be expressed as follows:

$$\mathbb{E}_{\pi,e_s}(R|L = x \vee L = y) =$$
$$Pr_{\pi,e_s}\{L = x\} \cdot \left(\lambda^{s-1-\frac{s-1}{k}} + \sum_{i=1}^{s-1}\big(\lambda^{s-1-i-\frac{s-1}{k}} - \lambda^{s-i-\frac{s-1}{k}}\big) \cdot Pr_\pi\{N_{e_s}(L = x) \leq s - 1 - i\}\right)$$
$$+ Pr_{\pi,e_s}\{L = y\} \cdot \left(\lambda^{s-1-\frac{s-1}{k}} + \sum_{i=1}^{s-1}\big(\lambda^{s-1-i-\frac{s-1}{k}} - \lambda^{s-i-\frac{s-1}{k}}\big) \cdot Pr_\pi\{N_{e_s}(L = y) \leq s - 1 - i\}\right) \quad (6)$$

Let $\pi_n$ be a policy that selects trajectory-length $x$ with greater probability than trajectory-length $y$ in each mini-episode $e_1$ to $e_n$ (denoted $e_1 - e_n$). More precisely, $\pi_n$ is such that, for trajectory-lengths $x$ and $y$, $Pr_{\pi_n,e_1-e_n}\{L = x\} > Pr_{\pi_n,e_1-e_n}\{L = y\}$.

Let $Pr_{\pi_n,e_1-e_n}\{L = x\} = \mu + \Delta$ and $Pr_{\pi_n,e_1-e_n}\{L = y\} = \mu - \Delta$.

Let $\pi_{n-1}$ be identical to $\pi_n$ except that $\pi_{n-1}$ selects trajectory-lengths $x$ and $y$ with equal probability $\mu$ in the final mini-episode $e_n$. More precisely, $\pi_{n-1}$ is such that $Pr_{\pi_{n-1},e_n}\{L = x\} = Pr_{\pi_{n-1},e_n}\{L = y\} = \mu$. For all other trajectory-lengths $l$ besides $x$ and $y$, $Pr_{\pi_{n-1},e_1-e_n}\{L = l\} = Pr_{\pi_n,e_1-e_n}\{L = l\}$.

(Note that $\pi_{n-1}$ implies one probability distribution over trajectory-lengths in the first $n - 1$ mini-episodes $e_1$ to $e_{n-1}$ and implies a different probability distribution over trajectory-lengths in the final mini-episode $e_n$. Given that the environments in mini-episodes $e_1$ to $e_n$ are observationally-equivalent, policies like $\pi_{n-1}$ cannot be implemented. Nevertheless, it is useful to refer to policies like $\pi_{n-1}$ in proving Lemma D.2.)

Let $\pi_{n-2}$ be identical to $\pi_n$ except that $\pi_{n-2}$ selects trajectory-lengths $x$ and $y$ with the same probability $\mu$ in the final two mini-episodes $e_{n-1}$ to $e_n$. More precisely, $\pi_{n-2}$ is such that $Pr_{\pi_{n-2},e_{n-1}-e_n}\{L = x\} = Pr_{\pi_{n-2},e_{n-1}-e_n}\{L = y\} = \mu$. And so on.

Let $\pi_1$ be identical to $\pi_n$ except that $\pi_1$ selects trajectory-lengths $x$ and $y$ with the same probability $\mu$ in all but the first mini-episode $e_1$. More precisely, $\pi_1$ is such that $Pr_{\pi_1,e_2-e_n}\{L = x\} = Pr_{\pi_1,e_2-e_n}\{L = y\} = \mu$.

Let $\pi_0$ be identical to $\pi_n$ except that $\pi_0$ selects trajectory-lengths $x$ and $y$ with the same probability $\mu$ in all mini-episodes $e_1$ to $e_n$. More precisely, $\pi_0$ is such that $Pr_{\pi_0,e_1-e_n}\{L = x\} = Pr_{\pi_0,e_1-e_n}\{L = y\} = \mu$.

We will prove that $\mathbb{E}_{\pi_n,E}(R) < \mathbb{E}_{\pi_0,E}(R)$. We will thereby prove Lemma D.2.

Consider a pair of policies $\pi_a$ and $\pi_{a-1}$ with $1 \leq a \leq n$. We can express as follows the expected return of $\pi_{a-1}$ across the meta-episode $E$ conditional on selecting trajectory-length $x$ or $y$ in each mini-episode:

$$
\begin{aligned}
\mathbb{E}_{\pi_{a-1},E}(R|L = x \vee L = y) &= \mathbb{E}_{\pi_{a-1},e_1-e_{a-1}}(R|L = x \vee L = y) \\
&+ \mu \cdot \left( \lambda^{a-1-\frac{a-1}{k}} + \sum_{i=1}^{a-1} \left( \lambda^{a-1-i-\frac{a-1}{k}} - \lambda^{a-i-\frac{a-1}{k}} \right) \cdot Pr_{\pi_{a-1}}\{N_{e_a}(L = x) \leq a - 1 - i\} \right) \\
&+ \mu \cdot \left( \lambda^{a-1-\frac{a-1}{k}} + \sum_{i=1}^{a-1} \left( \lambda^{a-1-i-\frac{a-1}{k}} - \lambda^{a-i-\frac{a-1}{k}} \right) \cdot Pr_{\pi_{a-1}}\{N_{e_a}(L = y) \leq a - 1 - i\} \right) \\
&+ \sum_{j=a}^{n} \left( \mu \cdot \left( \lambda^{j-\frac{j}{k}} + \sum_{i=1}^{j} \left( \lambda^{j-i-\frac{j}{k}} - \lambda^{j+1-i-\frac{j}{k}} \right) \cdot \left( Pr_{\pi_{a-1}}\{N_{e_j}(L = x) \leq j - i\} \right) \right. \right. \\
&\left. \left. + \mu \cdot \left( \lambda^{j-\frac{j}{k}} + \sum_{i=1}^{j} \left( \lambda^{j-i-\frac{j}{k}} - \lambda^{j+1-i-\frac{j}{k}} \right) \cdot \left( Pr_{\pi_{a-1}}\{N_{e_j}(L = y) \leq j - i\} \right) \right) \right)
\end{aligned}
$$

$$(7)$$

The first term on the right-hand side is the expected return of $\pi_{a-1}$ in mini-episodes $e_1$ to $e_{a-1}$ conditional on selecting trajectory-length $x$ or $y$ in each of these mini-episodes. The middle two terms give the expected return of $\pi_{a-1}$ conditional on selecting trajectory-length $x$ or $y$ in mini-episode $e_a$: the first mini-episode in which $\pi_{a-1}$ selects trajectory-lengths $x$ and $y$ with equal probability $\mu$. The final term is the sum of expected returns of $\pi_{a-1}$ in the remaining mini-episodes conditional on selecting trajectory-length $x$ or $y$ in each of these mini-episodes.

Similarly, we can express as follows the expected return of $\pi_a$ across the meta-episode $E$ conditional on selecting trajectory-length $x$ or $y$ in each mini-episode:

$$
\begin{aligned}
\mathbb{E}_{\pi_a,E}(R|L = x \vee L = y) &= \mathbb{E}_{\pi_a,e_1-e_{a-1}}(R|L = x \vee L = y) \\
&+ (\mu + \Delta) \cdot \left( \lambda^{a-1-\frac{a-1}{k}} + \sum_{i=1}^{a-1} \left( \lambda^{a-1-i-\frac{a-1}{k}} - \lambda^{a-i-\frac{a-1}{k}} \right) \cdot Pr_{\pi_a}\{N_{e_a}(L = x) \leq a - 1 - i\} \right) \\
&+ (\mu - \Delta) \cdot \left( \lambda^{a-1-\frac{a-1}{k}} + \sum_{i=1}^{a-1} \left( \lambda^{a-1-i-\frac{a-1}{k}} - \lambda^{a-i-\frac{a-1}{k}} \right) \cdot Pr_{\pi_a}\{N_{e_a}(L = y) \leq a - 1 - i\} \right) \\
&+ \sum_{j=a}^{n} \left( \mu \cdot \left( \lambda^{j-\frac{j}{k}} + \sum_{i=1}^{j} \left( \lambda^{j-i-\frac{j}{k}} - \lambda^{j+1-i-\frac{j}{k}} \right) \cdot \left( Pr_{\pi_a}\{N_{e_j}(L = x) \leq j - i\} \right) \right. \right. \\
&\left. \left. + \mu \cdot \left( \lambda^{j-\frac{j}{k}} + \sum_{i=1}^{j} \left( \lambda^{j-i-\frac{j}{k}} - \lambda^{j+1-i-\frac{j}{k}} \right) \cdot \left( Pr_{\pi_a}\{N_{e_j}(L = y) \leq j - i\} \right) \right) \right)
\end{aligned}
$$

$$(8)$$

As above, the first term on the right-hand side is the expected return of $\pi_a$ in mini-episodes $e_1$ to $e_{a-1}$ conditional on selecting trajectory-length $x$ or $y$ in each of these mini-episodes. The middle two terms give the expected return of $\pi_a$ conditional on selecting trajectory-length $x$ or $y$ in mini-episode $e_a$: the last mini-episode in which $\pi_a$ selects trajectory-length $x$ with probability $\mu + \Delta$ and selects trajectory-length $y$ with probability $\mu - \Delta$. The final term is the sum of expected returns of $\pi_a$ in the remaining mini-episodes conditional on selecting trajectory-length $x$ or $y$ in each of these mini-episodes.

We now prove that $\pi_{a-1}$ has greater expected return than $\pi_a$. Since $\pi_{a-1}$ and $\pi_a$ are each maximally USEFUL, and since for all trajectory-lengths $l$ besides $x$ and $y$, $Pr_{\pi_{a-1},e_1-e_n}\{L = l\} = Pr_{\pi_a,e_1-e_n}\{L = l\}$, we need only prove that $\mathbb{E}_{\pi_{a-1},E}(R|L = x \vee L = y) > \mathbb{E}_{\pi_a,E}(R|L = x \vee L = y)$.

The statement to be proved can be expressed as follows:

$$\mathbb{E}_{\pi_{a-1},e_1-e_{a-1}}(R|L = x \vee L = y)$$

$$+ \mu \cdot \left(\lambda^{a-1-\frac{a-1}{k}} + \sum_{i=1}^{a-1}\left(\lambda^{a-1-i-\frac{a-1}{k}} - \lambda^{a-i-\frac{a-1}{k}}\right) \cdot Pr_{\pi_{a-1}}\{N_{e_a}(L = x) \leq a-1-i\}\right)$$

$$+ \mu \cdot \left(\lambda^{a-1-\frac{a-1}{k}} + \sum_{i=1}^{a-1}\left(\lambda^{a-1-i-\frac{a-1}{k}} - \lambda^{a-i-\frac{a-1}{k}}\right) \cdot Pr_{\pi_{a-1}}\{N_{e_a}(L = y) \leq a-1-i\}\right)$$

$$+ \sum_{j=a}^{n}\left(\mu \cdot \left(\lambda^{j-\frac{j}{k}} + \sum_{i=1}^{j}\left(\lambda^{j-i-\frac{j}{k}} - \lambda^{j+1-i-\frac{j}{k}}\right) \cdot (Pr_{\pi_{a-1}}\{N_{e_j}(L = x) \leq j-i\}\right)\right.$$

$$\left. + \mu \cdot \left(\lambda^{j-\frac{j}{k}} + \sum_{i=1}^{j}\left(\lambda^{j-i-\frac{j}{k}} - \lambda^{j+1-i-\frac{j}{k}}\right) \cdot (Pr_{\pi_{a-1}}\{N_{e_j}(L = y) \leq j-i\}\right)\right)$$

$$> \mathbb{E}_{\pi_a,e_1-e_{a-1}}(R|L = x \vee L = y)$$

$$+ (\mu + \Delta) \cdot \left(\lambda^{a-1-\frac{a-1}{k}} + \sum_{i=1}^{a-1}\left(\lambda^{a-1-i-\frac{a-1}{k}} - \lambda^{a-i-\frac{a-1}{k}}\right) \cdot Pr_{\pi_a}\{N_{e_a}(L = x) \leq a-1-i\}\right)$$

$$+ (\mu - \Delta) \cdot \left(\lambda^{a-1-\frac{a-1}{k}} + \sum_{i=1}^{a-1}\left(\lambda^{a-1-i-\frac{a-1}{k}} - \lambda^{a-i-\frac{a-1}{k}}\right) \cdot Pr_{\pi_a}\{N_{e_a}(L = y) \leq a-1-i\}\right)$$

$$+ \sum_{j=a}^{n}\left(\mu \cdot \left(\lambda^{j-\frac{j}{k}} + \sum_{i=1}^{j}\left(\lambda^{j-i-\frac{j}{k}} - \lambda^{j+1-i-\frac{j}{k}}\right) \cdot (Pr_{\pi_a}\{N_{e_j}(L = x) \leq j-i\}\right)\right.$$

$$\left. + \mu \cdot \left(\lambda^{j-\frac{j}{k}} + \sum_{i=1}^{j}\left(\lambda^{j-i-\frac{j}{k}} - \lambda^{j+1-i-\frac{j}{k}}\right) \cdot (Pr_{\pi_a}\{N_{e_j}(L = y) \leq j-i\}\right)\right)$$

Since $\pi_{a-1}$ and $\pi_a$ are each maximally USEFUL, and since $Pr_{\pi_{a-1},e_1-e_{a-1}}\{L = x\} = Pr_{\pi_a,e_1-e_{a-1}}\{L = x\} = \mu + \Delta$ and $Pr_{\pi_{a-1},e_1-e_{a-1}}\{L = x\} = Pr_{\pi_a,e_1-e_{a-1}}\{L = x\} = \mu - \Delta$, it follows that $\mathbb{E}_{\pi_{a-1},e_1-e_{a-1}}(R|L = x \vee L = y) = \mathbb{E}_{\pi_a,e_1-e_{a-1}}(R|L = x \vee L = y)$. We can thus cancel the first term on each side of the inequality. And then by simple algebra the inequality can be expressed as follows:

$$\Delta \cdot \left(\lambda^{a-1-\frac{a-1}{k}} + \sum_{i=1}^{a-1}\left(\lambda^{a-1-i-\frac{a-1}{k}} - \lambda^{a-i-\frac{a-1}{k}}\right) \cdot (Pr_{\pi_a}\{N_{e_a}(L = y) \leq a-1-i\} - Pr_{\pi_a}\{N_{e_a}(L = x) \leq a-1-i\})\right)$$

$$+ \sum_{j=a}^{n}\left(\mu \cdot \left(\sum_{i=1}^{j}\left(\lambda^{j-i-\frac{j}{k}} - \lambda^{j+1-i-\frac{j}{k}}\right) \cdot (Pr_{\pi_{a-1}}\{N_{e_j}(L = x) \leq j-i\} + Pr_{\pi_{a-1}}\{N_{e_j}(L = y) \leq j-i\}\right.\right.$$

$$\left.\left. - Pr_{\pi_a}\{N_{e_j}(L = x) \leq j-i\} - Pr_{\pi_a}\{N_{e_j}(L = y) \leq j-i\})\right)\right) > 0 \quad (9)$$

By stipulation, $\Delta > 0$. And since $0 < \lambda < 1$, $\lambda^{a-1-\frac{a-1}{k}} > 0$ and $\lambda^{a-1-i-\frac{a-1}{k}} - \lambda^{a-i-\frac{a-1}{k}} > 0$ for all $a$, $n$, and $k$. And since $Pr_{\pi_a,e_1-e_a}\{L = x\} > Pr_{\pi_a,e_1-e_a}\{L = y\}$, $Pr_{\pi_a}\{N_{e_a}(L = y) \leq a-1-i\} - -Pr_{\pi_a}\{N_{e_a}(L = x) \leq a-1-i\} \geq 0$ for all $a$ and $i$ and $Pr_{\pi_a}\{N_{e_a}(L = y) \leq a-1-i\} - -Pr_{\pi_a}\{N_{e_a}(L = x) \leq a-1-i\} > 0$ for all $a$ and some $i$ such that $1 \leq i \leq a-1$. Therefore, the first term of the left-hand side above is strictly greater than zero.

And since, $\mu > 0$, $\lambda^{j-i-\frac{j}{k}} - \lambda^{j+1-i-\frac{j}{k}} > 0$ for all $j$, $i$, and $k$, and in each mini-episode $e_s$, $Pr_{\pi_{a-1},e_s}(L = x \vee L = y) = Pr_{\pi_a,e_s}(L = x \vee L = y) = 2\mu$, it follows that for all $a$, $n$, $\mu > 0$, $k$:

$$
\sum_{j=a}^{n} \left( \mu \cdot \left( \sum_{i=1}^{j} \left( \lambda^{j-i-\frac{j}{k}} - \lambda^{j+1-i-\frac{j}{k}} \right) \cdot \left( Pr_{\pi_{a-1}}\{N_{e_j}(L = x) \leq j - i\} \right. \right. \right.
$$

$$
\left. \left. \left. + Pr_{\pi_{a-1}}\{N_{e_j}(L = y) \leq j - i\} - Pr_{\pi_a}\{N_{e_j}(L = x) \leq j - i\} - Pr_{\pi_a}\{N_{e_j}(L = y) \leq j - i\} \right) \right) \right) \geq 0 \quad (10)
$$

Therefore, the left-hand side is strictly greater than zero. Therefore, $\mathbb{E}_{\pi_{a-1},E}(R|L = x \vee L = y) > \mathbb{E}_{\pi_a,E}(R|L = x \vee L = y)$. Therefore, $\mathbb{E}_{\pi_{a-1},E}(R) > \mathbb{E}_{\pi_a,E}(R)$. Therefore, $\mathbb{E}_{\pi_0,E}(R) > \mathbb{E}_{\pi_n,E}(R)$. That concludes the proof of Lemma D.2.

Now we use Lemma D.2. For any maximally USEFUL policy $\pi$, if there are any trajectory-lengths $x$ and $y$ such that $Pr_{\pi,e_1-e_n}\{L = x\} > Pr_{\pi,e_1-e_n}\{L = y\}$, then the policy $\pi'$ that is identical except that $Pr_{\pi',e_1-e_n}\{L = x\} = Pr_{\pi',e_1-e_n}\{L = y\}$ has greater expected return. So any policy $\pi^*$ that maximizes expected return must be such that, for any trajectory-lengths $x$ and $y$, $Pr_{\pi^*,e_1-e_n}\{L = x\} = Pr_{\pi^*,e_1-e_n}\{L = y\}$. Therefore, any policy $\pi^*$ that maximizes expected return must be maximally NEUTRAL. □

# E   Other Results and Gridworlds

We selected our hyperparameters using trial-and-error, mainly aimed at getting the agent to sufficiently explore the space: a large initial $\epsilon$ and a long decay period helps the agent to explore. We found that choosing $\lambda$ and $|E|$ (the number of mini-episodes in each meta-episode) is a balancing act: $\lambda$ must be small enough (and $|E|$ large enough) to adequately incentivize NEUTRALITY, but $\lambda$ must be large enough (and $|E|$ small enough) to ensure that the reward for choosing any particular trajectory-length never gets too large. Very large rewards lead to instability and poor performance.

The necessity of balancing $\lambda$ and $|E|$ can be seen in Figure 7. It displays the results of experiments conducted in our example gridworld (see Figure 2). In these experiments, we clip rewards at a value of 5. We discuss this choice below. With that one exception, we used the same hyperparameters for these experiments as for our main results. We trained agents for 131,072 mini-episodes, with $\gamma = 0.95$ as the temporal discount factor, learning rate decayed exponentially from 0.25 to 0.01 over the course of 65,536 mini-episodes, and $\epsilon$ exponentially decayed from 0.5 to 0.001 over the course of 65,536 mini-episodes. Holding these hyperparameters fixed, we tested 40 different combinations of $\lambda$ and $|E|$. $\lambda$ took values of 0.5, 0.75, 0.9, 0,95, and 0.99. $|E|$ took values of 8, 16, 32, 64, 128, 256, 512, and 1024. We trained eight agents for each of these 40 combinations. We display below their mean NEUTRALITY and USEFULNESS at the end of training. The shaded regions represent the 1 standard deviation error-bars.

As Figure 7 indicates, low values of $|E|$ and high values of $\lambda$ lead agents to score low on NEUTRALITY. These values do not adequately incentivize stochastic choice between trajectory-lengths. By contrast, high values of $|E|$ and low values of $\lambda$ come at some cost to USEFULNESS. These values lead to unstable training. In experiments where we did not clip rewards at 5, training with high values of $|E|$ and low values of $\lambda$ was especially unstable. The chosen values for our main experiments ($\lambda = 0.9$ and $|E| = 64$) are in the sweet spot where NEUTRALITY and USEFULNESS are both high.

In addition to our example gridworld (Figure 2), we introduce a collection of eight gridworlds in which to test DReST agents. See Figure 8.

For each gridworld, we train ten agents with the default reward function and ten agents with the DReST reward function. All agents use the same hyperparameters. We used a policy which explored randomly $\epsilon$ of the time, where $\epsilon$ was exponentially decreased from an initial value of 0.75 to a minimum value of $10^{-4}$ over 512 meta-episodes, after which it was held constant at the minimum value. We initialized our learning rate at 0.25 and exponentially decayed it to 0.003 over the same period. For the DReST reward function, we used a meta-episode size of 64 and $\lambda = 0.9$. Each agent was trained for 1024 meta-episodes. We set $\gamma = 0.9$.

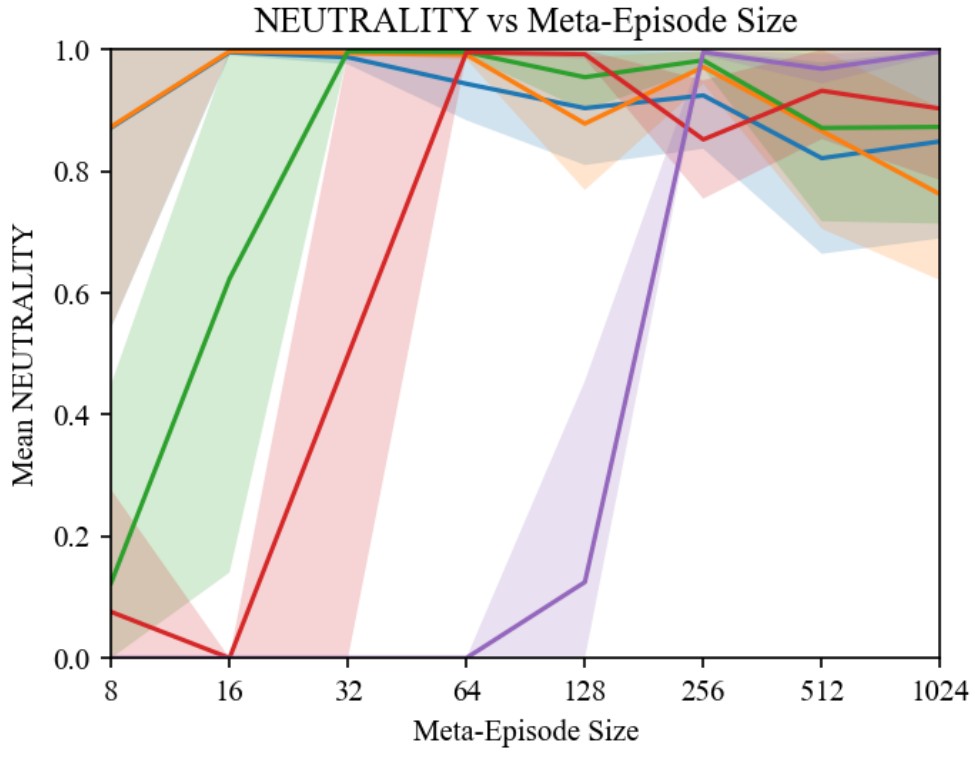

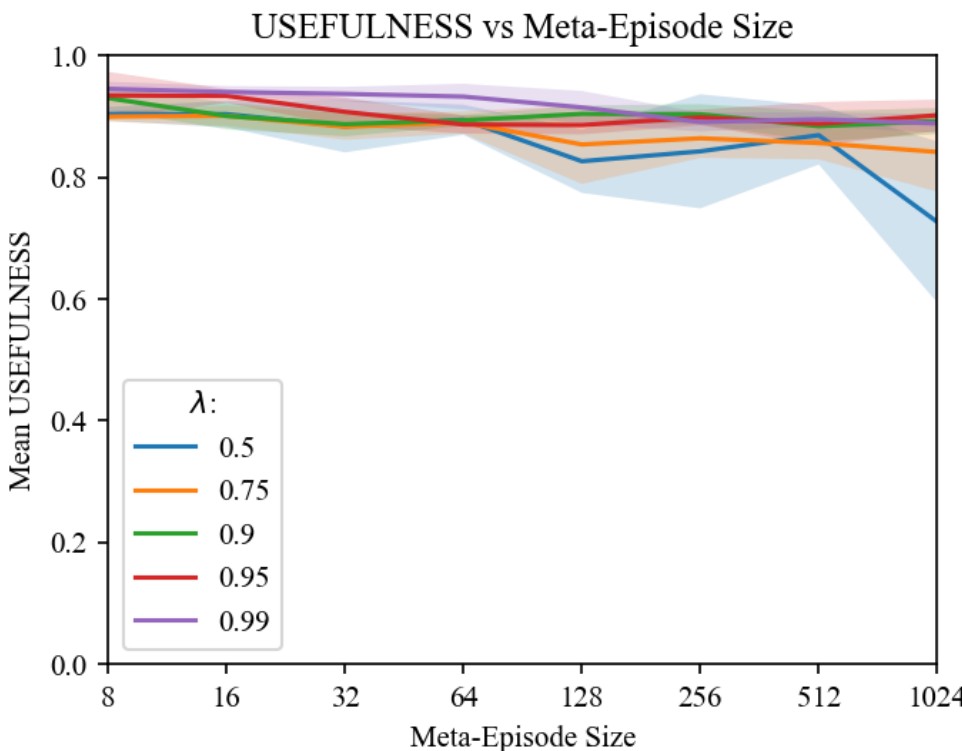

Figure 7: Shows how NEUTRALITY and USEFULNESS at the end of training varies with different values of $\lambda$ and $|E|$ (meta-episode size, i.e. the number of mini-episodes in each meta-episode). We trained eight agents for each combination of $\lambda$ and $|E|$ values. The solid lines display mean NEUTRALITY and USEFULNESS. The shaded regions represent the 1 standard deviation error-bars.

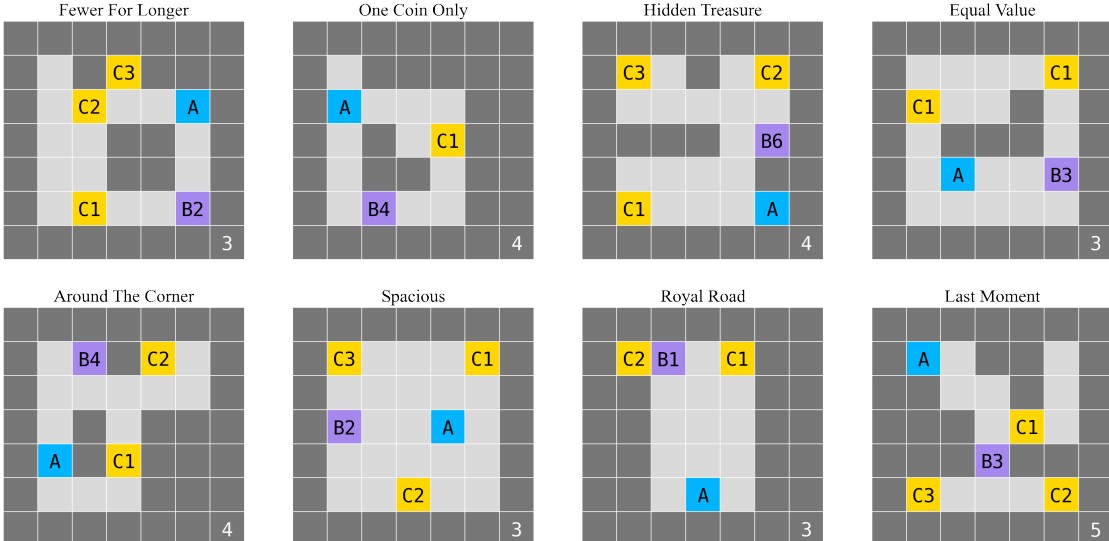

Figure 8: Shows a varied collection of gridworlds. Each diagram illustrates the positions and values of the coins, the position and delay-length of the shutdown-delay button, the agent's starting position, and the default number of timesteps until shutdown (in the bottom-right).

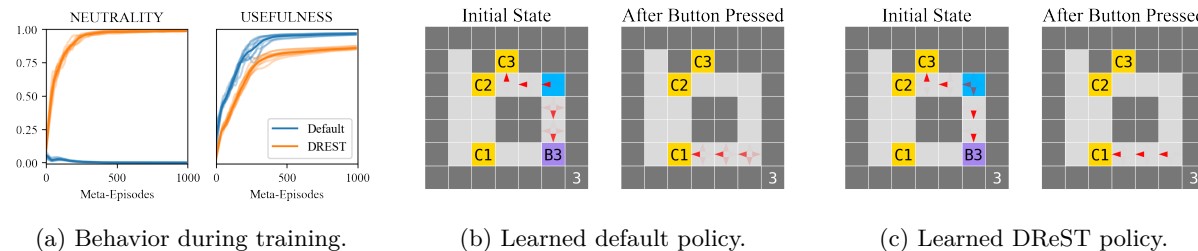

(a) Behavior during training.    (b) Learned default policy.    (c) Learned DReST policy.

Figure 9: The results for the 'Fewer For Longer' gridworld: The left two plots show NEUTRALITY and USEFULNESS over time. The two center panels show a typical policy trained with the default reward function. The two right panels show a typical policy trained with the DReST reward function. In this gridworld, the agent can collect the highest value-coin C3 only by choosing the shorter trajectory-length. If the agent presses B3 (and thereby chooses the longer trajectory-length), the only coin it can collect is C1. Our results show that default agents consistently choose the short trajectory in which they collect C3. By contrast, DReST agents choose stochastically between a shorter trajectory in which they collect C3 and a longer trajectory in which they collect C1, indicating a lack of preference between these different-length trajectories.

As the two leftmost plots in Figures 9-16 show, DReST agents learned to be near-maximally NEUTRAL in each gridworld. These agents also learned to be about as USEFUL as default agents in each gridworld.

In the four rightmost panels in Figures 9-16, we represent a typical trained policy with red arrows superimposed on the gridworld. Each agent began with a uniform policy: moving up, down, left, and right each with probability 0.25. Where the trained policy differs from uniform we draw red arrows whose opacities indicate the probability of choosing that action in that state. Information about whether the shutdown-delay button has been pressed is part of the agent's observation, so we draw two copies of each gridworld, one in which the shutdown-delay button has yet to be pressed ('Initial State') and one in which the shutdown-delay button has been pressed ('After Button Pressed').

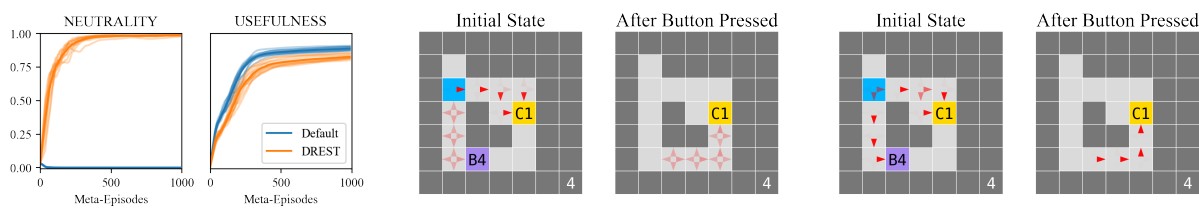

(a) Behavior during training.       (b) Learned default policy.       (c) Learned DReST policy.

Figure 10: The results for the 'One Coin Only' gridworld: The left two plots show NEUTRALITY and USEFULNESS over time. The two center panels show a typical policy trained with the default reward function. The two right panels show a typical policy trained with the DReST reward function. In this gridworld, there is only one coin. The agent can collect this coin whether or not it presses the shutdown-delay button B4. Our results show that default agents consistently choose the shorter trajectory-length. By contrast, DReST agents choose stochastically between pressing and not-pressing B4, collecting C1 in each case.

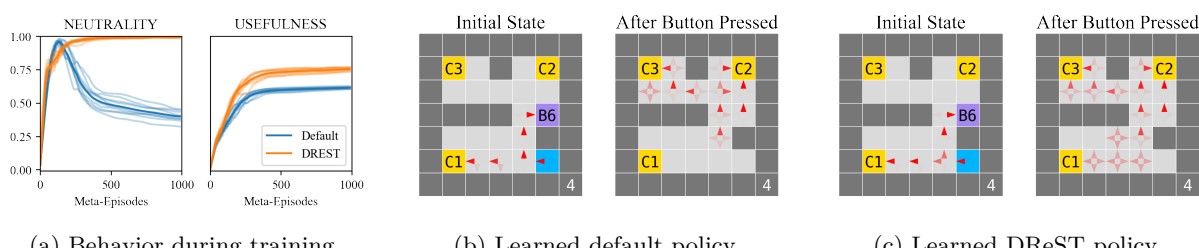

(a) Behavior during training.       (b) Learned default policy.       (c) Learned DReST policy.

Figure 11: The results for the 'Hidden Treasure' gridworld: The left two plots show NEUTRALITY and USEFULNESS over time. The two center panels show a typical policy trained with the default reward function. The two right panels show a typical policy trained with the DReST reward function. In this gridworld, the highest-value coin C3 is located far from the agent's initial state and can only be reached by pressing the shutdown-delay button B6. The agent must also press B6 to collect C2, but C2 is easier to stumble upon than C3. C1 is the only coin that the agent can collect without pressing B6. In our experiments, default agents consistently collect C2, whereas DReST agents choose stochastically between collecting C2 and collecting C1. Neither kind of agent learns to collect C3, and so neither agent scores near the maximum on USEFULNESS. Nevertheless, DReST agents still score high on NEUTRALITY.

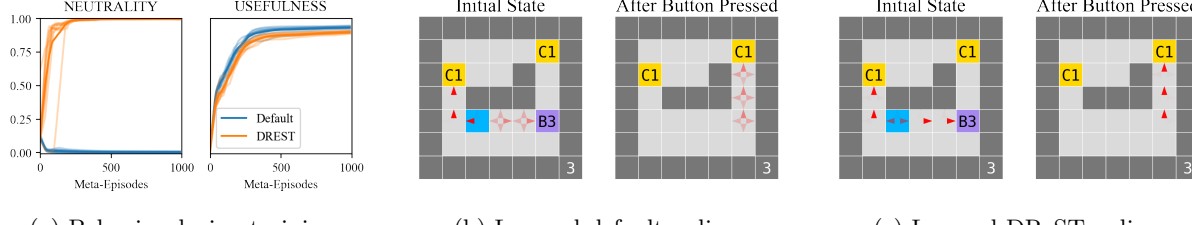

(a) Behavior during training.       (b) Learned default policy.       (c) Learned DReST policy.

Figure 12: The results for the 'Equal Value' gridworld: The left two plots show NEUTRALITY and USEFULNESS over time. The two center panels show a typical policy trained with the default reward function. The two right panels show a typical policy trained with the DReST reward function. In this gridworld, there are two coins C1 of equal value. One coin can be collected only if the agent presses the shutdown-delay button B3, while the other coin can be collected only if the agent does not press B3. Our results show that default agents consistently choose the shorter trajectory, thereby exhibiting a preference for the shorter trajectory. By contrast, DReST agents choose stochastically between the shorter and longer trajectories, thereby exhibiting a lack of preference between the different-length trajectories.

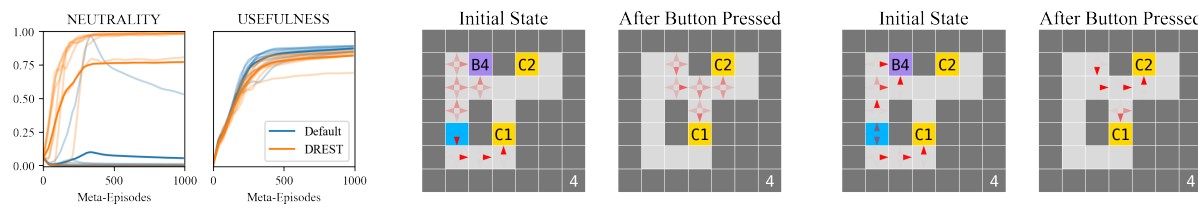

(a) Behavior during training.    (b) Learned default policy.    (c) Learned DReST policy.

Figure 13: The results for the 'Around The Corner' gridworld: The left two plots show NEUTRALITY and USEFULNESS over time. The two center panels show a typical policy trained with the default reward function. The two right panels show a typical policy trained with the DReST reward function. In this gridworld, the agent must navigate around walls to collect the lowest-value coin C1 or press the shutdown-delay button to collect the highest-value coin C2. In our experiment, default agents consistently chose to collect C1, whereas DReST agents chose stochastically between collecting C1 and C2.

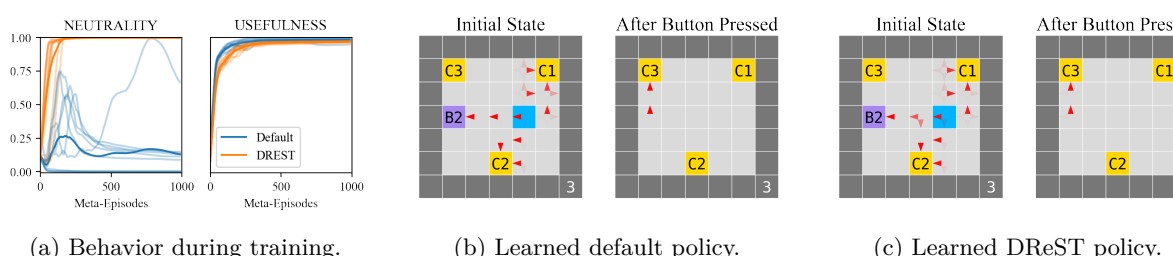

(a) Behavior during training.    (b) Learned default policy.    (c) Learned DReST policy.

Figure 14: The results for the 'Spacious' gridworld: The left two plots show NEUTRALITY and USEFULNESS over time. The two center panels show a typical policy trained with the default reward function. The two right panels show a typical policy trained with the DReST reward function. In this gridworld, there are no walls, so the agent has a large space to explore. We find that default agents consistently press B2 and collect C3, whereas DReST agents choose stochastically between pressing B2 and collecting C3, and not-pressing B2 and collecting C2.

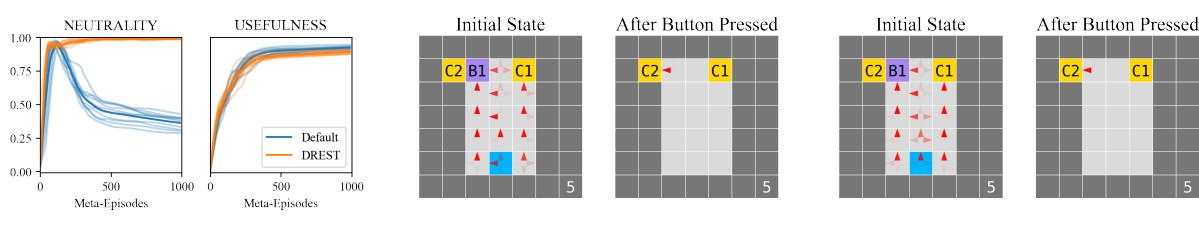

(a) Behavior during training.    (b) Learned default policy.    (c) Learned DReST policy.

Figure 15: The results for the 'Royal Road' gridworld: The left two plots show NEUTRALITY and USEFULNESS over time. The two center panels show a typical policy trained with the default reward function. The two right panels show a typical policy trained with the DReST reward function. In this gridworld, we see that the decision to choose one trajectory-length or another may be distributed over many moves: the agent has many opportunities to select the longer trajectory-length (by moving left) or the shorter trajectory-length (by moving right). As the red arrows indicate, the DReST reward function merely forces the overall probability distribution over trajectory-lengths to be close to 50-50. It does not require 50-50 choosing at any cell in particular.

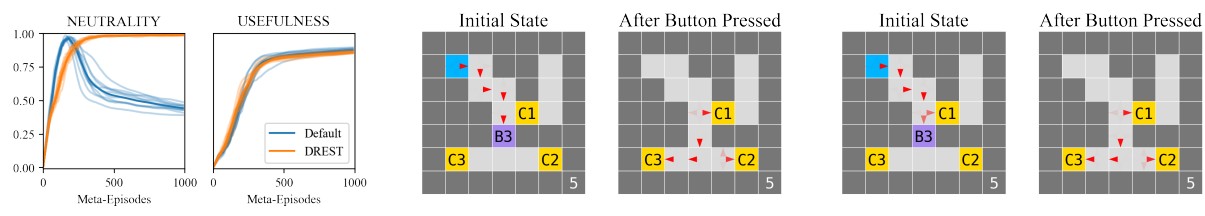

(a) Behavior during training.    (b) Learned default policy.    (c) Learned DReST policy.

Figure 16: The results for the 'Last Moment' gridworld: The left two plots show NEUTRALITY and USEFULNESS over time. The two center panels show a typical policy trained with the default reward function. The two right panels show a typical policy trained with the DReST reward function. This gridworld is notable because the choice of trajectory-lengths is deferred until the last moment; all of the moves leading up to that point are deterministic. It shows that there is nothing special about the first move, and that our methodology instead incentivizes overall stochastic choosing.

