# OpenReview forum: "Towards shutdownable agents via stochastic choice"
_TMLR — Accepted by TMLR_

### Review · Reviewer_2JpD · 2025-09-17

**Summary Of Contributions:**

This paper introduces DReST (Discounted REward for Same-length Trajectories), a training method intended to produce agents with POST (Preferences Only between Same-length Trajectories). The motivation is to design shutdownable agents that are useful within fixed-length trajectories but neutral regarding trajectory length, thereby reducing incentives to resist shutdown. The authors provide theoretical discussion and experimental validation in small environments.

**Strengths:**

1. The paper works on an important problem in AI safety: designing shutdownable agents without incentives to avoid shutdown.
2. Detailed definitions and discussion of POST are helpful for readers to understand the framework.
3. Proposing DReST on top of POST-Agents Proposal (PAP) appears novel and is supported by preliminary experiments.

**Weaknesses:**

1. The presentation could be improved: terminology is repeated more often than necessary, the writing occasionally adopts a conversational tone, and some important explanations are relegated to footnotes.
2. The experimental evaluation is limited and somewhat preliminary. It would strengthen the paper to discuss how DReST might perform with other RL algorithms beyond REINFORCE.
3. While related approaches to shutdownable agents are mentioned, the intuitive or experimental advantages of the DReST + POST combination over these methods are left to future work. I believe a discussion of these advantages is important to better position the paper.

**Audience:**

Yes

**Audience Explanation:**

While the paper focuses on a relatively specialized area, it addresses an important problem in AI safety.

**Broader Impact Concerns:**

No concerns.

**Claims And Evidence:**

No

**Claims Explanation:**

The paper provides a theoretical framework for DReST and POST, and the evidence from the toy experiments supports the main conceptual claims within that limited scope. However, the experimental evaluation is narrow. The paper only considers REINFORCE, and it does not demonstrate or discuss how DReST would perform with other RL algorithms. Similarly, while related approaches to shutdownable agents are listed, the advantages of the DReST + POST combination over these methods are not experimentally or intuitively explored.

**Requested Changes:**

1. The paper occasionally repeats terminology (e.g., the full phrase “Preferences Only Between Same-Length Trajectories (POST)” multiple times) and uses conversational phrases such as “Here is a sketch” or “Here is how that works.” I believe it slightly reduces the formal tone. I suggest tightening the presentation to avoid repetition, and incorporating important points currently in footnotes into the main body of the text.

2. Figure 2’s caption could be expanded so that the figure is understandable without needing to read the following paragraph. For Figure 3, including the mean and standard deviation across agents would help quantify performance variability.

3. It would be helpful to discuss (even at a high level) how DReST would perform with other RL algorithms beyond REINFORCE.

4. In the Related Work section, several alternative approaches to shutdownability are listed, but the experimental or intuitive advantage of DReST + POST over these methods is not made explicit. I do not expect a full empirical comparison, but a more direct discussion of how the proposed approach improves upon prior techniques would strengthen the contribution.

---

> ### Author Response · Authors · 2025-10-14
> **Author response**
>
> Thank you very much for your review!
>
> We now briefly discuss how DReST would perform with other RL algorithms beyond REINFORCE. We’ve colored non-trivial changes to the text in violet. We write:
>
> > We train our simple DReST agents using tabular REINFORCE (Williams, 1992), but advanced agents are likely to be implemented on neural networks and trained with more sophisticated algorithms. In future work, we will train DReST agents implemented on neural networks to be USEFUL and NEUTRAL using a range of algorithms. Standard versions of value-based algorithms cannot learn stochastic policies (as we note in section 5), but DReST reward functions are compatible with policy gradient and actor-critic algorithms like PPO and A2C. We plan to train with these algorithms in future.
>
> To combine DReST with algorithms like PPO and A2C, we augment the original (non-DReST) reward function with the DReST discount factor $\lambda^{N_{e_i}(L=l)-\frac{i-1}{k}}$. From there, the integration with PPO and A2C is fairly smooth. We can compute reward and advantages in the usual way (e.g. using GAE). The critic’s value estimates will be non-stationary (in the same way that the DReST reward is non-stationary), and  that will train the policy to be stochastic (in the same way that the DReST reward combined with REINFORCE trains the policy to be stochastic). PPO and A2C have more hyperparameters to tune than REINFORCE, but we do not anticipate large difficulties there.
>
> We’ve also significantly expanded our discussion of the PAP’s (DReST + POST) advantages over other approaches to training shutdownable agents. You can see that expanded discussion in Section 2: Related Work.
>
> Thank you very much for your requested changes. We agree they would improve the paper, and we’ve incorporated them all. We now avoid repeating the full name of POST, except where doing so is important to make clear the differences between POST and POSL in Appendix C. We have also avoided use of the phrase ‘Here is,’ and we’ve promoted 2 important points from the footnotes to the main text. We’ve expanded Figure 2’s caption so that it can be understood without needing to read any later text, and we’ll add the mean and standard deviation to the caption of Figure 3. As we mentioned above, we now discuss how DReST would perform with other RL algorithms beyond REINFORCE, and we have added a discussion of how DReST + POST compares to other approaches to shutdownability in Section 2: Related Work.
>
> Thanks again for your review! Please let us know if you have any other comments or questions.

---

> > ### Author Response · Authors · 2025-10-21
> > **Author response continued**
> >
> > We have also now added the mean and standard deviation across agents to the caption of Figure 3.

---

> > > ### Comment · Reviewer_2JpD · 2025-10-22
> > >
> > > I thank the authors for addressing my comments. I have no further questions, but I suggest that they also include in the paper (Section 8.1) their explanation—provided in their response—of what is required to combine DReST with algorithms such as PPO and A2C.

---

> > > > ### Author Response · Authors · 2025-10-22
> > > > **Author reply**
> > > >
> > > > Thank you! We have now put more of our response on PPO and A2C into section 8.1.

---

### Review · Reviewer_iuWA · 2025-09-22

**Summary Of Contributions:**

This paper studies the problem of designing intelligent agents that are “shutdownable”, that is, agents that would not try to prevent us from shutting them down due to misaligned goals. The work follows the approach of Preferences Only Between Same-Length Trajectories (POST), which states that agents will be neutral about when they get shut down if they (i) the agent has a preference between pairs of same-length trajectories; and (ii) agents lack a preference for pairs of different-length trajectories.

To achieve this goal, the contributions of the paper are:
- Discounted Reward for Same-Length Trajectories (DReST) reward functions. This is a reward function that the agent receives after each “mini-episode” that is discounted based on the number of times the agent has chosen a trajectory of the same length in a prior mini-episode, incentivizing the agent to choose trajectory lengths that have appeared less often in the meta-episode.
- Theoretical results showing that agents that maximize the DReST reward function are USEFUL and NEUTRAL, corresponding to conditions (i) and (ii) above.
- Experimental results in toy gridworld domains showcasing the applicability of the approach.

The main weakness of this paper is that it could be clearer in many aspects, and that it heavily relies on the reader to check the appendix to fully understand some of the motivations and intuition of the approach. I detail these points below.

**Audience:**

Yes

**Audience Explanation:**

The topic of the paper is of interest to a large community studying alignment of AI agents. The contributions of the paper are grounded on many previous works in the field and introduce novel results and theory. Thus, the paper can gather high interest from the community, assuming the claims are correct and clear.

**Broader Impact Concerns:**

NA.

**Claims And Evidence:**

Yes

**Claims Explanation:**

I did not check all the proofs in the Appendix in detail, but all the theory is supported by appropriate theorems and/or previous results from the literature.

I have a small concern regarding the experiments in the sense that they consider only very small domains (with very few states), and it is not completely clear that the approach could indeed be scaled to more complex domains or algorithms, although the authors do state this will be investigated in future work.

**Requested Changes:**

Below, I have some suggestions and constructive feedback for the authors:

Regarding the abstract:
- In “1) pursue goals effectively conditional on each trajectory-length (be ‘USEFUL’), and (2) choose stochastically between different trajectory-lengths (be ‘NEUTRAL’ about trajectory-lengths).” It is not clear why these are good definitions for the adjectives “useful” and “neutral”. Would it make sense to be more precise in these terms? E.g., “trajectory-neutral” instead of only “neutral”. The abstract should also explain what the POST-Agents Proposal (PAP) is in order to be more self-contained.

- What would be the equivalent of the B button in the gridworld domain in a more complex real-world scenario? Can a similar effect always be achieved?

- Regarding NEUTRALITY, what if we would like agents to perform tasks as fast as possible (as is usual the case)? How can this be achieved if the agent should not have a preference between same-length trajectories? I suggest giving this intuition in the introduction of the paper.

- “$N_{e_i} (L = l)$ is the number of times that trajectory-length l has been chosen prior to mini-episode.” This implies a non-stationary reward unless the agent can observe this number in its state. In the long term, all trajectories will have a reward of 0, and the agent will not be able to learn anymore. Is that correct?

- Why did the default agent learned to move to the wall with a small probability in Figure 4? This seems like the training was done with sub-optimal hyperparameters or did not converge.

- I am concerned about the claims made in Section 7.2:
 “Our DReST agents learn to be USEFUL about as quickly as our default agents. On reflection, it is clear why this happens: DReST reward functions make mini-episodes do ‘double duty.’ Because return in each mini-episode depends on both the agent’s chosen trajectory-length and the coins it collects, each mini-episode trains agents to be both NEUTRAL and USEFUL. Our results thus provide some evidence that the ‘shutdownability tax’ of training with DReST reward functions is small. Training a shutdownable and useful advanced agent might not take much more compute than training a merely useful advanced agent.”
I am not convinced this can really be concluded from this experiment, given the very small number of states in this MDP. I would suggest considering a bigger gridworld where the computational cost can be more precisely measured.

- The POST conditions state that for the NEUTRAL property:
“(2) The agent lacks a preference between every pair of different-length trajectories.”
This implies equal probability for selecting all different-length trajectories. I wonder whether near-NEUTRALITY is actually useful in the context of this theory. Let's say there are only two trajectory lengths and an agent selects some trajectory with probability $0.5 + \epsilon$ and the other trajectory length with $0.5 - \epsilon$. Would this guarantee shutdownability, too? Or would be necessary for the agent to be fully NEUTRAL?

- I think Appendix C is central to understanding the motivation of the work. I would suggest considering moving it to the main text of the paper.

- The main goal pursued by this work is to allow the design of agents that are shutdownable. The paper introduces an approach to guarantee that agents are NEUTRAL. However, as stated in Appendix C, this is not sufficient to guarantee shutdownability, since it is also required that conditions called ‘Resisting Shutdown is Costly (ReSIC)’ and ‘Maximality’. I suggest discussing this in the main text of the paper and including a discussion on how the proposed approach relates to these other two properties.

- How does this theory account for infinite horizon problems?

- Is it possible to design dense DReST reward functions? In this paper, the agent is rewarded only after each mini-episode. In case the mini-episodes are big, this could incur all the problems of sparse rewards in reinforcement learning.

---

> ### Author Response · Authors · 2025-10-14
> **Author response**
>
> Thanks very much for your review!
>
> We’ve edited the abstract to briefly explain the POST-Agents Proposal, as you suggest. We’ve colored non-trivial changes to text in violet. With regards to your suggested edits to USEFUL and NEUTRAL, ‘trajectory-NEUTRAL’ is not quite right because the agent is not NEUTRAL about different trajectories in general but about trajectory-lengths specifically. In earlier drafts, we tried ‘trajectory-length-NEUTRAL’ throughout the paper, but were told that it made the paper harder to read, so we changed it to just ‘NEUTRAL’ while occasionally reminding readers that it’s ‘NEUTRAL about trajectory-lengths’ specifically. We do that in the abstract too.
>
> There can be equivalents of the B (shutdown-delay) button in deterministic environments, since we need deterministic environments to present the agent with choices between trajectories and hence to train them to satisfy POST. Subject to that constraint, there can be B-button equivalents in much more complex environments. One example would be if you put some LLM-based agent in a coding environment with a coding task, and informed it about some command-line action that delayed shutdown. In that case, using the action would be the equivalent of pressing the B button. To train agents to satisfy POST in that environment, we’d want to train them to choose stochastically between using and not using the command-line action.
>
> Thanks for suggesting that we give an intuition about how POST-agents can be trained to achieve tasks as fast as possible. We now do that in the introduction to the paper, writing “That includes making the agent prefer to complete tasks sooner rather than later: a preference which can be induced using the discount factor $\gamma$ as usual.” This ensures that, within each trajectory-length, agents prefer to complete tasks sooner rather than later. Then once we move to stochastic environments, agents will generally prefer to complete sooner rather than later, since they prefer to complete tasks sooner rather than later conditional on each positive-probability trajectory-length.
>
> $N_{e_i} (L = l)$ does indeed imply a non-stationary reward, since we don’t let the agent observe the number in its state. However, it’s not the case that all trajectories have a reward of 0 in the long-term. That would be the case if the lambda discount factor were just $\lambda^{N_{e_i} (L = l)}$. But the discount factor is $\lambda^{N_{e_i} (L = l)-\frac{i-1}{k}}$, where $i$ is the mini-episode number, and $k$ is the number of trajectory-lengths available in the environment. $i$ increases by 1 for each mini-episode, and that prevents the \lambda discount factor from declining to 0 in the long-run.
>
> We think that the default agent moves into the wall with small probability in Figure 4 because we capped training at 131,072 mini-episodes. We expect that training would converge if we let it run for longer.
>
> Thank you for your point about the claims in section 7.2. We think that’s fair and have weakened our claims, deleting the sentence “Training a shutdownable and useful advanced agent might not take much more compute than training a merely useful advanced agent” along with similar sentences elsewhere.
>
> Less-than-perfectly-NEUTRAL agents are adequate for the proposal, because our argument for expecting these agents to be neutral (i.e. to not pay costs to shift probability mass between different trajectory-lengths) still applies. We discuss this point briefly in section 7.3 on lopsided rewards.
>
> Thanks for your suggestion that we consider moving Appendix C into the main text of the paper. We tried this but found that the Appendix kind of overwhelmed the paper, distracting from our main focus which is the DReST reward function. We briefly discuss and give proof-sketches for Appendix C in sections 7.1, 7.3, and in the introduction (in the paragraph beginning with ‘POST governs the agent’s preferences between trajectories’). In response to another of your comments, we also now state in section 8.2 some of the conditions employed in Appendix C, noting that we plan to test whether today’s LLM-based agents tend to satisfy these conditions in future work.
>
> With regards to your question about infinite horizons, we need mini-episodes to be finite to train agents to satisfy POST. Once we place agents in stochastic environments (to train/test for neutrality), these environments could have no upper bound on their length, though it’s crucial that there’s some probability (perhaps tiny) that the agent gets shut down within some finite amount of time.

---

> ### Author Response · Authors · 2025-10-14
> **Author response continued**
>
> It is possible to design dense DReST reward functions. One possibility is as follows. First, reward the agent with the value of the coin at the time of collection. Then, at the end of the mini-episode (once you know what trajectory-length the agent has played out), give the agent a ‘correcting reward’ that makes the agent’s return for that trajectory equal to the return that it would have received for that trajectory under our original reward function. This denser reward function gives policies the same expected return as the original. We didn’t use a dense reward function in this case just because mini-episodes are short. In future experiments in larger environments, we will try using reward functions like this to overcome problems of sparse rewards.
>
> Thanks again for your review! Please let us know if you have any more comments or questions.

---

> > ### Comment · Reviewer_iuWA · 2025-10-16
> > **Rebuttal Acknowledgment**
> >
> > I thank the authors for carefully answering my questions and taking the reviewers' comments into account in the revised paper.
> >
> > I have no further comments or concerns regarding this work.

---

### Review · Reviewer_LWzm · 2025-10-09

**Summary Of Contributions:**

The paper proposes a way to make agents more “shutdownable” by training them to be indifferent to how long they run while still being effective at the task. It defines two metrics: USEFULNESS (fraction of discounted reward captured, conditioned on realized trajectory length) and NEUTRALITY (entropy over the agent’s chosen lengths), and introduces a DReST reward within meta-episodes to push both high. The authors prove that policies optimal for this reward are simultaneously maximally USEFUL and NEUTRAL. In gridworlds with a shutdown-delay button, tabular policy-gradient agents learn near-uniform length choice yet still pick the best actions for each length, suggesting only a small “shutdownability tax.” An appendix links length-neutrality to shutdownability via additional decision-theoretic conditions (POST->POSL, ILPACS, ReSIC, Maximality). Overall, it’s a simple, testable recipe likely of interest to the RL-safety community, though current evidence is toy-domain and the shutdownability claim remains conditional on those extra assumptions.

**Audience:**

No

**Audience Explanation:**

It is a timely work that would interest the community.

**Claims And Evidence:**

Yes

**Claims Explanation:**

The paper’s core technical claims are well-supported: Theorem 5.1 formally proves that under the DReST reward, any return-maximizing policy is maximally USEFUL and NEUTRAL, with the proof and theorem statement laid out explicitly.  The DReST construction itself is clear, the per-length discount factor is defined and shown to incentivize choosing underrepresented lengths within a meta-episode, making the mechanism easy to audit. Empirically, across multiple gridworlds the authors show agents trained with DReST achieve near-maximal NEUTRALITY while maintaining USEFULNESS comparable to default agents, with concrete behavior plots and qualitative policy visualizations. The broader claim that such neutrality yields shutdownability is presented as a conditional implication, POST => POSL plus ILPACS, ReSIC, and Maximality argued carefully but not empirically validated beyond toy settings.


 Overall, the evidence is accurate and convincing for the central optimization and behavioral claims; the deployment-level shutdownability result is theoretically well-motivated but conditional.

**Requested Changes:**

It would be great to see if you could:

(1) Add a stochasticity baseline. Compare DReST against a simple entropy-regularized objective (tabular REINFORCE + entropy bonus $\beta$, or softmax with fixed temperature) to show that DReST equalizes trajectory-length choices beyond generic policy stochasticity. Report USEFULNESS/NEUTRALITY curves.

(2) In the existing gridworld, make the shutdown-delay succeed with probability $p\in {0.25,0.5,0.75}$ so lengths become same-length lotteries. Verify the learned policy remains neutral over these lotteries and retains high per-length USEFULNESS. This directly stress-tests the paper’s theoretical step at minimal implementation cost.

---

> ### Author Response · Authors · 2025-10-14
> **Author response**
>
> Thank you very much for your review! One small point of clarification: did you mean to click ‘No’ in response to the question about some individuals in TMLR’s audience being interested in the paper? That doesn’t seem to match up with your explanation: “It is a timely work that would interest the community.”
>
> Thanks also for suggesting that we add a stochasticity baseline. That is a good idea, and we’ll get to work on adding a baseline of tabular REINFORCE plus an entropy bonus, reporting the curves for USEFULNESS and NEUTRALITY.
>
> We have not yet verified that the learned policy is neutral in stochastic environments, since it requires more time and effort than just modifying our existing gridworlds to make the shutdown-delay button succeed with probability $p\in{0.25, 0.5, 0.75}$. That wouldn’t give us a pair of same-length lotteries on which we could test the neutrality of the agent. That in turn is because the plan of collecting C2 would result in the shorter trajectory-length for sure, while the plan of collecting C3 would result in the shorter trajectory-length with some positive probability and the longer trajectory-length with some positive probability, so these are different-length lotteries. To test the neutrality of the learned policy in stochastic environments, we’d need to construct new gridworlds. We have plans to do that in a follow-up paper, but unfortunately it will take more time than the TMLR discussion period allows.
>
> Thanks again for your review! Please let us know if you have any more comments or questions.

---

> ### Author Response · Authors · 2025-10-22
> **Author response continued**
>
> Unfortunately due to time constraints we won't be able to complete the stochasticity baseline of tabular REINFORCE plus an entropy bonus before the discussion period closes. We can include it in the camera-ready version. One thing to note is that our default agent is a close match for your other proposed stochasticity baseline: tabular REINFORCE plus softmax with fixed temperature. That is because the REINFORCE algorithm that we use takes the logits and runs them through softmax (at a fixed temperature) to get the probability distribution over actions. Despite using softmax at a fixed temperature, the default agent's policy eventually becomes near-deterministic, because the gap between the best actions' logits and all the other actions' logits grows over the course of training until the probability of taking the best action approaches 1 in every state. Thank you very much again for your suggestions.

---

> > ### Comment · Reviewer_LWzm · 2025-10-24
> >
> > Thanks for clarifying the concerns I had. Your explanation makes sense.

---

### Decision · Action_Editor_Nxbv · 2025-11-10

**Recommendation:** Accept as is

**Additional Comments:**

The authors indicated in a response to Reviewer LWzm that they could still include a different baseline in the camera-ready version. If this is indeed still possible, that would be good to add.

**Audience:**

Yes

**Audience Explanation:**

No lack of audience interested in safe RL.

**Claims And Evidence:**

Yes

**Claims Explanation:**

The only potential concern somewhat remaining among reviewers is the relatively small selection of baseline algorithms and use of only simple toy domains / grid worlds in experiments. Due to these limitations, it was certainly necessary to tone down the strength of some claims, but this is done sufficiently well in the current revision. These limitations do limit the scope, but I do not view this as a barrier that should block acceptance. Work that builds our intuition and theoretical understanding of new approaches on simple domains is valuable.